# Mixing times of organic molecules within secondary organic aerosol particles: a global planetary boundary layer perspective

Adrian M. Maclean[1], Christopher L. Butenhoff[2], James W. Grayson[1], Kelley Barsanti[3], Jose L. Jimenez[4*], Allan K. Bertram[1*]

[1]Department of Chemistry, University of British Columbia, Vancouver, BC, V6T 1Z1, Canada
[2]Dept. of Physics, Portland State University, Portland, Oregon
[3]Department of Chemical and Environmental Engineering and Center for Environmental Research and Technology, University of California, Riverside
[4]Cooperative Institute for Research in the Environmental Sciences and Department of Chemistry and Biochemistry, University of Colorado, Boulder, CO, USA

*To whom correspondence should be addressed. Allan K. Bertram (email): bertram@chem.ubc.ca and Jose L. Jimenez (email): jose.jimenez@colorado.edu

**Abstract**

When simulating the formation and life cycle of secondary organic aerosol (SOA) with chemical transport models, it is often assumed that organic molecules are well mixed within SOA particles on the time scale of 1 h. While this assumption has been debated vigorously in the literature, the issue remains unresolved in part due to a lack of information on the mixing times within SOA particles as a function of both temperature and relative humidity. Using

laboratory data, meteorological fields, and a chemical transport model, we estimated how often mixing times are < 1 h within SOA in the planetary boundary layer (PBL), the region of the atmosphere where SOA concentrations are on average the highest. First, a parameterization for viscosity as a function of temperature and RH was developed for α-pinene SOA using room-temperature and low-temperature viscosity data for α-pinene SOA generated in the laboratory using mass concentrations of ~1000 μg m$^{-3}$. Based on this parameterization, the mixing times within α-pinene SOA

are < 1 h for 98.5 % and 99.9 % of the occurrences in the PBL during January and July, respectively, when concentrations are significant (total organic aerosol concentrations are > 0.5 μg m$^{-3}$ at the surface). Next, as a starting point to quantify how often mixing times of organic molecules are < 1 h within α-pinene SOA generated using low, atmospherically relevant, mass concentrations, we developed a temperature-independent parameterization for viscosity using the room-temperature viscosity data for α-pinene SOA generated in the laboratory using a mass

concentration of ~70 μg m$^{-3}$. Based on this temperature-independent parameterization, mixing times within α-pinene SOA are < 1 h for 45 and 38 % of the occurrences in the PBL during January and July, respectively, when concentrations are significant. However, associated with these conclusions are several caveats, and due to these caveats, we are unable to make strong conclusions about how often mixing times of organic molecules are < 1 h within α-pinene SOA generated using low, atmospherically relevant, mass concentrations. Finally, a parameterization for

viscosity of anthropogenic SOA as a function of temperature and RH was developed using sucrose-water data. Based on this parameterization, and assuming sucrose is a good proxy for anthropogenic SOA, 70 % and 83 % of the mixing

times within anthropogenic SOA in the PBL are < 1 h for January and July, respectively, when concentrations are significant. These percentages are likely lower limits due to the assumptions used to calculate mixing times.

## 1. Introduction

Secondary organic aerosol (SOA) is formed in the atmosphere when volatile organic compounds from biogenic and anthropogenic sources are oxidized by a complex series of reactions to form semivolatile organic compounds (SVOCs), followed by condensation of the lowest volatility products or reactions of the SVOCs in the particle phase (Ervens et al., 2011; Hallquist et al., 2009). The term "secondary" indicates the aerosol is formed in the atmosphere rather than emitted directly into the atmosphere in the particle phase. Globally, SOA from biogenic sources dominate, with SOA from anthropogenic sources contributing approximately 10 % to the total SOA budget (Hallquist et al., 2009; Spracklen et al., 2011). Major contributors to biogenic SOA are oxidation products of α-pinene and isoprene (Hu et al., 2015; Kanakidou et al., 2005; Pathak et al., 2007), and as a result, SOA derived from α-pinene and isoprene are the most widely used representatives of biogenic SOA in experimental and modelling studies.

The planetary boundary layer (PBL) is the lowest part of the atmosphere, ranging from the Earth's surface to roughly 1 km in altitude, depending on location and time (Wallace and Hobbs, 2006). Within this region vertical mixing of air masses is rapid, on the order of 30 minutes (Wallace and Hobbs, 2006). In addition, within the PBL the temperature varies from roughly 265 K to 305 K and the relative humidity (RH) varies from roughly 20 % to 100 % (see below). SOA concentrations are also on average highest in the PBL (Heald et al., 2011; Wagner et al., 2015).

When simulating the formation, growth, and evaporation of SOA particles with chemical transport models, it is often assumed that SVOCs are well mixed within SOA particles on the time scale of 1 h (Hallquist et al., 2009). If SVOCs are not well mixed within SOA particles on this time scale, then chemical transport models could incorrectly predict SOA mass concentrations by up to an order of magnitude (Shiraiwa and Seinfeld, 2012) and incorrectly predict the size of SOA particles (Zaveri et al., 2014), with implications for air quality and climate predictions (Seinfeld and Pandis, 2006). Recent research has shown that mixing times of organic molecules within SOA particles can be > 1 h at room-temperature and low RHs (Abramson et al., 2013; Grayson et al., 2016; Liu et al., 2016; Perraud et al., 2012; Renbaum-Wolff et al., 2013; Song et al., 2016; Ye et al., 2016; Zhang et al., 2015). In addition, studies have shown that proxies of SOA particles can form glasses at low RHs and low-temperatures (Koop et al., 2011; Zobrist et al., 2008). Nevertheless, the conditions that lead to slow mixing times in SOA may be infrequent on a global scale in the PBL. If this is the case, then the assumption of well mixed SOA particles in chemical transport models should be reasonable. How often mixing times are > 1 h under ambient conditions in the PBL is not well constrained, in part due to the lack of information on mixing times of organic molecules in SOA particles as a function of both RH and temperature.

In the following, we a) developed a parameterization for the viscosity of α-pinene SOA particles as a function of both RH and temperature, b) determined the distribution of RH and temperature in the PBL from an archive of meteorological fields, c) determined the conditions in the PBL when SOA concentrations are significant using a chemical transport model, and d) quantified how often mixing times of SVOCs are > 1 h within α-pinene for ambient

temperatures and RHs in the PBL. Mixing times within anthropogenic SOA and the effect of SOA mass concentration on mixing times are also discussed.

## 2. Materials and Methods

### 2.1 Parameterization for the viscosity of α-pinene SOA as a function of temperature and RH

The following data was used to develop a parameterization of the viscosity α-pinene SOA as a function of temperature and RH: a) room-temperature measurements of viscosity of SOA derived from α-pinene ozonolysis by Grayson et al. (2016) (Table S1), b) low-temperature measurements of viscosity for SOA derived from α-pinene ozonolysis by Järvinen et al. (2016) (Table S2), and c) temperature dependent measurements of viscosity for water from Crittenden et al. (2012) (Table S3). Järvinen et al. (2016) measured the temperature and RH values at which α-pinene SOA has

a viscosity of approximately $10^7$ Pa s. In these experiments, SOA was generated with a mass concentration of 707-1414 µg m$^{-3}$. Grayson et al. (2016) measured viscosity of α-pinene SOA as a function of RH at 295 K. In these experiments, the SOA was generated with mass concentrations of 121 µg m$^{-3}$ and 520 µg m$^{-3}$. We use the viscosity measurements from Grayson et al. (2016) determined with a mass concentration of 520 µg m$^{-3}$ to be more consistent with the mass concentrations used by Järvinen et al. (2016). Although there are other room-temperature measurements

of the viscosity of α-pinene SOA (Bateman et al., 2015; Hosny et al., 2016; Kidd et al., 2014; Pajunoja et al., 2014; Renbaum-Wolff et al., 2013), we used the room-temperature measurements from Grayson et al. (2016) because 1) viscosity was measured over a range of relative humidities in this study, 2) the mass concentrations used by Grayson et al. (2016) to generate the SOA were similar to the mass concentrations used by Järvinen et al. (2016), and 3) Grayson et al. (2016) measured the viscosity of the total SOA (both the water soluble component and the water insoluble

component).

Due to the experimental conditions used by Grayson et al. (2016) and Järvinen et al. (2016), the parameterization developed here is applicable to SOA generated using a mass concentration of ~ 1000 µg m$^{-3}$. We focused on ~ 1000 µg m$^{-3}$ because both low-temperature and room-temperature viscosity measurements have been carried out using this mass concentration. The effect of mass concentration on the viscosity α-pinene SOA is discussed in Section 3.5.

To develop a parameterization for viscosity as function of temperature and RH, the following equation was fit to the measurements by Grayson et al. (2016), Järvinen et al. (2016), and Crittenden et al. (2012) (Table S1-S3):

$$\log(\eta) = 12 - \frac{C_1 * (T - \frac{w_{SOA}T_{gSOA} + w_{H2O}T_{gH2O}k_{GT}}{w_{SOA} + w_{H2O}k_{GT}})}{C_2 + (T - \frac{w_{SOA}T_{gSOA} + w_{H2O}T_{gH2O}k_{GT}}{w_{SOA} + w_{H2O}k_{GT}})} \tag{1}$$

where $C_1$ and $C_2$ are constants, $k_{GT}$ is the Gordon-Taylor fitting parameter, $T_{gSOA}$ and $T_{gH2O}$ are the glass transition temperatures of dry SOA and water, and $w_{SOA}$ and $w_{H2O}$ are the weight fractions of the dry SOA and water in the

particles. The weight fractions of the dry SOA and water in the particles were determined from the RH using the following equation (Koop et al., 2011):

$$\frac{RH}{100} = \frac{1}{(1 + i_{SOA}\frac{n_{SOA}}{n_{H2O}})} \tag{2}$$

where i is the van't Hoff factor and n is the number of moles of dry SOA and water in the particles. We assumed a value of 1 for the van't Hoff factor (Koop et al., 2011) and a dry molecular weight for SOA of 175 g mol[-1] (Huff Hartz et al., 2005).

Since the glass transition temperature of water is known (135 K) (Corti et al., 2008), the unknowns in Eq. (1) (and hence fitting parameters) were $C_1$, $C_2$, $k_{gt}$ and $T_{gSOA}$. The values for these parameters retrieved by fitting the equation to the viscosity data discussed above (using a non-linear curve fitting function in Matlab) are reported in Table S4.

Equation (1) was based on the Williams, Landel, and Ferry (WLF) equation and the Gordon-Taylor equation. The WLF equation provides a relationship between viscosity and temperature:

$$\log(\frac{\eta}{\eta_g}) = \frac{-C_1(T-T_g)}{C_2+(T-T_g)}$$ (3)

where $C_1$ and $C_2$ are constants, T is the temperature, $T_g$ is the glass transition temperature, $\eta$ is the viscosity and $\eta_g$ is the viscosity at the glass transition ($10^{12}$ Pa s). The Gordon-Taylor equation provides a relationship between the glass transition temperature of a mixture and the weight fractions of its components:

$$T_{g,mix} = \frac{w_1 T_{g1} + w_2 T_{g2} k_{GT}}{w_1 + w_2 k_{GT}}$$ (4)

where $w_1$ and $w_2$ are the weight fractions of the solute and water, $T_{g1}$ and $T_{g2}$ are the glass transition temperatures of the solute and water, and $k_{GT}$ is a fitting parameter that describes the interaction between the two species. Equations (3) and (4) can be combined to give Eq. (1).

Equation (3) (and hence Eq. (1)) is valid only at or above the glass transition temperature. As a result, we have not used Eq. (1) to predict viscosities > $10^{12}$ Pa s (which corresponds to mixing times > $5 \times 10^5$ h). If the temperature and RH in the PBL were such that the viscosity was greater than $10^{12}$ Pa s, we assigned a viscosity of $10^{12}$ Pa s and a mixing time of $5 \times 10^5$ hours. This assignment does not affect the conclusions in this manuscript since a mixing time of $5 \times 10^5$ hours is already well above the residence time of SOA particles in the atmosphere.

## 2.2. Organic aerosol concentrations in the planetary boundary layer

To determine the conditions in the PBL when SOA concentrations are significant we used the global chemical transport model GEOS-Chem (http://acmg.seas.harvard.edu/geos/). The version of GEOS-Chem used (v10-01) includes organic aerosol (OA) formation from semi-volatile and intermediate volatility organic compounds (SVOC and IVOC) (Pye and Seinfeld, 2010), plus new aerosol production from nitrate radical oxidation of isoprene and terpenes and NO$_x$-dependent aerosol yields from terpenes (Pye et al., 2010). In this version IVOC emissions are spatially distributed based on naphthalene. To estimate SVOC emissions we scaled the default GEOS-Chem primary organic aerosol (POA) emissions inventory by 1.27 following Pye and Seinfeld (2010). GEOS-Chem was run at a horizontal grid resolution of 4° latitude by 5° longitude using GEOS-5 meteorology with 47 vertical layers with a 3-year spin-up period. Shown in Fig. 1 are the monthly averaged total organic aerosol concentrations at the surface for the months of January and July 2006. These monthly averaged total organic aerosol concentrations were used to

remove times and locations where SOA concentrations are not expected to be of major importance for climate, health or visibility.

**2.3 RH and temperature in the PBL**

Information on the RH and temperature distributions in the global PBL in different seasons were also needed to assess mixing times within SOA particles. First, the afternoon PBL heights were determined globally using the 6-h averaged GEOS-5 meteorology fields. Then, temperature and RH in each grid cell within the PBL were determined globally using the 6-h averaged GEOS-5 meteorology fields. To determine if a grid cell was within the PBL, the afternoon PBL heights mentioned above were used. The GEOS-5 archive provides temperature and RH at a horizontal grid resolution of 4° latitude by 5° longitude and 47 vertical layers.

**3. Results and Discussion**

**3.1 Parameterization of viscosity and mixing times within α-pinene SOA particles as a function of RH and temperature**

Shown in Fig. 2a (contours) is the RH and temperature dependent parameterization for α-pinene SOA viscosities based on viscosities measured at room-temperature (Grayson et al., 2016) and low-temperature (Järvinen et al., 2016), as well as the viscosity of water as a function of temperature (Crittenden et al., 2012). From the viscosity parameterization, the diffusion coefficients of organic molecules within α-pinene SOA particles were calculated using the Stokes-Einstein equation:

$$D = \frac{kT}{6\pi\eta R_H} \tag{5}$$

where $D$ is the diffusion coefficient, $k$ is the Boltzmann constant, $T$ is temperature in Kelvin, $\eta$ is the dynamic viscosity and $R_H$ is the hydrodynamic radius of the diffusing species. For the calculations, a hydrodynamic radius of 0.38 nm was used for the diffusing organic molecules within SOA, based on an assumed molecular weight of 175 g mol$^{-1}$ (Huff Hartz et al., 2005), a density of 1.3 g cm$^{-3}$ (Chen and Hopke, 2009; Saathoff et al., 2009) and spherical symmetry. The Stokes-Einstein equation should give reasonable values when the radius of the diffusing molecules is roughly greater than or equal to the radius of the matrix molecules and when the viscosity of the matrix is relatively small ($\lesssim$ 400 Pa s) (Chenyakin et al., 2017; Price et al., 2016). When the viscosity of the matrix is large ($\gtrsim 10^6$ Pa s), the Stokes-Einstein equation can underpredict diffusion coefficients of organic molecules in organic matrices (Champion et al., 1997; Chenyakin et al., 2017; Price et al., 2016). Hence, the diffusion coefficients and mixing times estimated here should be considered lower and upper limits, respectively.

From the diffusion coefficients, the mixing times of organic molecules within an α-pinene SOA particle were calculated with the following equation (Shiraiwa et al., 2011):

$$\tau_{mix} = \frac{d^2}{4\pi^2 D} \tag{6}$$

where $\tau_{mix}$ is the mixing time, d is the diameter of an SOA particle, and D is the diffusion coefficient estimated from Eq. (5). For these calculations, it was assumed that the α-pinene SOA particles have a diameter of 200 nm, which is

roughly the median diameter in the volume distribution of ambient SOA-containing particles (Martin et al., 2010; Pöschl et al., 2010; Riipinen et al., 2011). Once the mixing time has elapsed, the concentration of the diffusing molecules at the centre of the particle is within 1/e of the equilibrium concentration (Shiraiwa et al., 2011). The calculated mixing times (Fig. 2b) illustrate that, as expected, inverse relationships exist between both mixing time and

RH, as well as mixing time and temperature.

**3.2 RH and temperature in the PBL**

Shown in Figs. 3a and 3b are the normalized frequency counts of temperature and RH in the PBL for the months of January and July, 2006, respectively, based on the archive of meteorological fields (GEOS-5) used in the global chemical transport model, GEOS-Chem, v10-01. We only included grid points in our analysis if the grid points were

within the PBL and the monthly average mass concentration of total organic aerosol was > 0.5 μg m$^{-3}$ at the surface, based on GEOS-Chem, v10-01 (Fig. 1). In other words, we included all the grid points in a column up to the top of the PBL when determining frequency counts if the monthly averaged total organic aerosol concentration was > 0.5 μg m$^{-3}$ at the surface. This filtering removes cases where SOA concentrations are not expected to be of major importance for climate, health or visibility. We chose a mass concentration of > 0.5 μg m$^{-3}$ for filtering because the mass

concentration of total organic aerosol at the surface was > 0.5 μg m$^{-3}$ in all but one of the previous field measurements of organic aerosol at remote locations (Spracklen et al., 2011).

The normalized frequency counts illustrate that the temperature and RH in the PBL are often in the range of 290-300 K and > 50 % RH for the month of January (Fig. 3a) and in the range of 285-300 K and > 30 % RH for the month of

July (Fig. 3b). For reference, shown in Figs. S1 and S2 are the average temperature and RH conditions at the Earth's surface and top of the planetary boundary layer, respectively, for January and July, based on the archive of meteorological fields for 2006 (GEOS-5).

**3.3 Mixing times of organic molecules within α-pinene SOA particles the PBL**

Also shown in Figs. 3a and 3b are the mixing times within 200 nm α-pinene SOA particles predicted with our

parameterization (contours). These results, together with the frequency counts of temperature and RH throughout the vertical column of the PBL, indicate that the mixing times of organic molecules within α-pinene SOA are often < 1x10$^{-1}$ h for conditions in the PBL.

Shown in Fig. 4 are the normalized frequency distributions of mixing times within α-pinene SOA for January and July, based on the data in Figs. 3a and 3b. Figure 4 suggests that the mixing times within α-pinene SOA are < 1 h for

98.5 % and 99.9 % of the occurrences in the PBL during January and July, respectively, when monthly average total organic aerosol concentrations are > 0.5 μg m$^{-3}$ at the surface.

Within the PBL, RH increases and temperature decreases with altitude, with both changes being substantial and impacting mixing times in opposite directions. Shown in Fig. 5a-c are calculated monthly average afternoon (13:00-15:00, local time) vertical profiles of temperature, RH, and mixing times within α-pinene SOA over Hyytiälä (boreal

forest), and the Amazon (rainforest) for the driest month of the year at each location (the method used to calculate

vertical profiles is described in the Supporting Information, Section S1). Afternoon vertical profiles were chosen since this is the time of the day when RH is typically lowest and thus mixing times are the longest. Figure 5c shows that mixing times within α-pinene SOA decrease significantly with altitude for these two locations. This is because the plasticizing effect of water on viscosity dominates the temperature effect for these conditions.

Shown in Fig. 6 are global maps of the monthly averaged mixing times of organic molecules within α-pinene SOA for conditions at the top of the PBL for the months of January and July. Figure 6 shows that 91.2 % and 97.5 % of the locations for January and July, respectively, have a mixing time < 0.1 h for conditions at the top of the PBL when monthly averaged total organic aerosol surface concentrations are > 0.5 µg m$^{-3}$. Within the PBL, vertical mixing of air masses occurs on the order of 30 min. Since the mixing times within α-pinene SOA particles for conditions at the

top of the PBL are < 0.1 h for most locations where the SOA concentrations are significant (total organic aerosol concentration > 0.5 µg m$^{-3}$ at the surface), a reasonable upper limit to the mixing time within the α-pinene SOA studied here for most locations in the PBL is 30 min. During this 30 min interval, mixing times within α-pinene SOA particles can cycle between short and long values, though rarely being > 1 h (Figs. 3 and 4).

**3.4 Sensitivity analysis**

To calculate the mixing times discussed above, we assumed that the α-pinene SOA particles have a diameter of 200 nm. We also repeated these calculations assuming a diameter of 500 nm, since aged organic aerosol can have larger diameters (Takegawa et al., 2006). Based on the viscosity parameterization shown in Fig. 2a, mixing times within 500 nm α-pinene SOA particles are < 1 h for 95.9 % and 99.4 % of the occurrences in the PBL during January and July, respectively (Fig. S3).

The parameterization of viscosity used above was developed using viscosity measurements by Grayson et al. (2016), Järvinen et al. (2016) and Crittenden et al. (2012). As a sensitivity analysis, we developed a second parameterization, using the same procedure as describe above, but using the upper limits to the viscosities reported by Grayson et al. (2016) and the upper limits to the RH ranges reported by Järvinen et al. (2016). This should result in an upper limit to

the viscosity parametertization discussed above. The uncertainties in the measurements by Crittenden et al. (2012) were not considered since they are small compared to the uncertainties reported by Grayson et al. (2016) and Järvinen et al. (2016). Based on this second parameteriation, mixing times are < 1 h for 96.6 % and 99.5 % of the occurrences in the PBL during January and July, respectively, when the total organic aerosol was > 0.5 µg m$^{-3}$ at the surface (Fig. S4).

**3.5 Effect of mass concentration used to generate the SOA**

The parameterizations developed above were based on SOA generated using a mass concentration of ~ 1000 µg m$^{-3}$. As mentioned, we focused on ~ 1000 µg m$^{-3}$ because low-temperature and room-temperature viscosity measurements have been carried out using this mass concentration. However, the viscosity of some types of SOA may depend on the mass concentration used to generate the SOA. For example, Grayson et al. (2016) showed that under dry conditions,

the viscosity of α-pinene SOA may increase by a factor of 5 as the production mass concentration decreased from

1200 µg m⁻³ to 120 µg m⁻³. In addition, mass concentrations of biogenic SOA are typically $\leq 10$ µg m⁻³ in the atmosphere (Spracklen et al., 2011). As a starting point to quantify how often mixing times of organic molecules are < 1 h within α-pinene SOA generated using low, atmospherically relevant, mass concentrations, we developed a temperature-independent parameterization using the room-temperature viscosity data for α-pinene SOA from Zhang

et al. (2015) (Table S5) and room-temperature viscosity data for water from (Crittenden et al., 2012) (Table S3). Zhang et al. (2015) measured the viscosity of α-pinene SOA over a range of relative humidities (0-60 %), and the SOA used in these experiments was generated in the laboratory using a mass concentration of ~70 µg m⁻³. The median room-temperature viscosities reported by Zhang et al. are higher than the median room-temperature viscosities reported by Grayson et al. (2016) using a mass concentration of 520 µg m⁻³ (Fig. S5). Although not proven, a

reasonable explanation for the difference in median viscosities is the difference in mass concentrations used to generate the SOA.

A temperature-independent parameterization was generated by fitting Eq. (1) to the room-temperature viscosity data from Zhang et al. (2015) and Crittenden et al. (2012), but with the temperature (T) in Eq. (1) replaced by 293 K. The values for the parameters retrieved by fitting the modified Eq. (1) to the viscosity data are reported in Table S6. The

temperature-independent parameterization generated using this method is shown in Fig.7a. Shown in Figs. 7b, 8a, and 8b (contours) is the parameterization for mixing times within 200 nm α-pinene SOA based on this temperature-independent viscosity parameterization. Also included in Figs. 8a and 8b are the normalized frequency counts of temperature and RH in the PBL for the months of January and July, 2006, respectively, when the monthly average mass concentration of total organic aerosol was > 0.5 µg m⁻³ at the surface. Shown in Fig. 9 are the normalized

frequency distributions of mixing times within α-pinene SOA for January and July, based on the data in Figs. 8a and 8b. Figure 9 suggests that the mixing times within α-pinene SOA is < 1 h for 45 and 38 % of the occurrences in the PBL during January and July, respectively, when monthly average total organic aerosol concentrations were > 0.5 µg m⁻³ at the surface. However, several caveats need to be emphasized: 1) the parameterization was developed based on room-temperature viscosity data only. Viscosities, and hence mixing times, will increase as temperature decreases.

As an illustration, the viscosity of sucrose-water mixtures can increase by 2-3 orders of magnitude as the temperature decreases by 10 K close to the glass transition temperature (Champion et al., 1997). 2) The mixing times were calculated using the Stokes-Einstein relation, which can underpredict diffusion coefficients, and hence overpredict mixing times, when the viscosity of the matrix is high. For example, the Stokes-Einstein equation can underpredict diffusion coefficients of organic molecules in sucrose-water mixtures by at least a factor of 10 to 100 at viscosities

$\geq 10^6$ Pa s (Chenyakin et al., 2017; Price et al., 2016). 3) The viscosity data from Zhang et al. (2015) has an uncertainty of $\pm 1$ order of magnitude, which was not considered in the temperature-independent parameterization. Considering these caveats, we are unable to make strong conclusions about how often mixing times of organic molecules are < 1 h within α-pinene SOA generated at low mass concentrations. To help resolve this issue, temperature dependent studies of the viscosity of α-pinene SOA generated using low mass concentrations are needed.

Also, the accuracy of the Stokes-Einstein equation for predicting diffusion coefficients of organics within α-pinene SOA needs to be determined.

**3.6 Mixing times of organic molecules within anthropogenic SOA particles in the PBL**

Recently it has been shown that the diffusion rates of organics in SOA from toluene photooxidation are slower than the diffusion rates of organics in SOA from α-pinene ozonolysis at room-temperature (Liu et al., 2016; Song et al., 2016; Ye et al., 2016). These results indicate that that mixing times are longer in some types of anthropogenic SOA than some types of biogenic SOA, at least at room-temperature. SOA derived from anthropogenic sources can be a significant contributor to SOA over polluted regions (Hallquist et al., 2009; Spracklen et al., 2011). Viscosities or diffusion rates within toluene SOA or other types of anthropogenic SOA have yet to be measured at temperatures lower than room-temperature. As a result, we have used sucrose as a proxy of anthropogenic SOA, since the viscosity of sucrose is similar to the viscosity of toluene SOA at room-temperature (Fig. S6) (Power and Reid, 2014; Song et al., 2016), and since a parameterization of the viscosity of sucrose as a function of temperature and RH can be developed using literature data. In the Supporting Information (Section S2, Table S7-S9, and Figs. S7-S10) we carried out a similar analysis for sucrose as for α-pinene SOA above. Assuming sucrose is a good proxy for anthropogenic SOA, the analysis suggests that 70 % and 83 % of the mixing times within anthropogenic SOA in the PBL are < 1 h for January and July, respectively, when SOA concentrations are significant (total organic aerosol concentration > 0.5 μg m$^{-3}$ at the surface). In addition, 81 % and 87 % of the locations for January and July, respectively, have a mixing time < 0.1 h at the top of the PBL when surface concentrations of total organic aerosol are > 0.5 μg m$^{-3}$. These percentages for anthropogenic SOA are likely lower limits since, as mentioned earlier, studies have shown that the Stokes-Einstein relation (which is used here to calculate diffusion coefficients of organic molecules from viscosities) can underpredict diffusion coefficients of organic molecules in sucrose-water mixtures by at least a factor of 10 to 100 at viscosities $\geq 10^6$ Pa s (Chenyakin et al., 2017; Price et al., 2016). Measurements of diffusion rates of organic molecules within anthropogenic SOA as a function of both temperature and RH are needed to better constrain how often mixing times are > 1 h within anthropogenic SOA in the PBL.

**3.7 Comparison with previous studies**

Shiraiwa et al. (2017) recently estimated mixing times of organics within SOA in the troposphere using a global chemistry climate model and a relationship between glass transition temperatures, molar mass, and oxygen-to-carbon elemental ratios. Their results suggest mixing times of organics within SOA are short (< 1 min) over the oceans, tropics, and high latitudes at the surface and 850 hPa. On the other hand, their results suggest mixing times are long (> 1 hour) over dry regions (i.e. major deserts) at the surface and at 850 hPa and over most continental regions at 850 hPa. The general trends observed by Shiraiwa et al. (2017) are consistent with the trends observed here. However, the mixing times predicted by Shiraiwa et al. (2017) appear to be longer than the mixing times predicted here using viscosities of α-pinene SOA generated with a mass concentration ∼ 1000 μg m$^{-3}$. Quantitative differences between the current work and the work by Shiraiwa et al. (2017) are not surprising since Shiraiwa et al. (2017) considered both anthropogeneic SOA and biogenic SOA simultaneously, and since they used a very different approach to estimate viscosities of atmospheric SOA.

**4. Summary and Conclusions**

A parameterization for viscosity as a function of temperature and RH was developed for α-pinene SOA based on room-temperature and low-temperature viscosity data of α-pinene SOA generated in the laboratory using mass concentrations of ~1000 μg m$^{-3}$. We focused on ~1000 μg m$^{-3}$ because low-temperature and room-temperature

viscosity measurements have been carried out using this mass concentration. Based on this parameterization, as well as RH and temperatures in the PBL, the mixing times within α-pinene SOA are < 1 h for 98.5 % and 99.9 % of the occurrences in the PBL during January and July, respectively, when monthly average total organic aerosol concentrations are > 0.5 μg m$^{-3}$ at the surface. Also based on this parameterization, 91.2 % and 97.5 % of the locations for January and July, respectively, have a mixing time < 0.1 h for conditions at the top of the PBL when monthly

averaged total organic aerosol surface concentrations are > 0.5 μg m$^{-3}$.

As a starting point to quantify how often mixing times of organic molecules are < 1 h within α-pinene SOA generated using low mass concentrations, we developed a temperature-independent parameterization using the room-temperature viscosity data for α-pinene SOA from Zhang et al. (2015). Zhang et al. (2015) measured the viscosity of α-pinene SOA generated using a mass concentration of ~70 μg m$^{-3}$. Based on this temperature-independent

parameterization, mixing times within α-pinene SOA are < 1 h for 45 and 38 % of the occurrences in the PBL during January and July, respectively, when monthly average total organic aerosol concentrations are > 0.5 μg m$^{-3}$ at the surface. However, several caveats need to be emphasized for these results. Most important, the results were based on room-temperature viscosity data only and the mixing times were calculated using the Stokes-Einstein relation, which can underpredict diffusion coefficients of organic molecules, and hence overpredict mixing times, when the viscosity

of the matrix is high.

As a starting point to quantify how often mixing times of organic molecules are < 1 h within anthropogenic SOA, a parameterization for viscosity as a function of temperature and RH was developed using sucrose-water viscosity data. Based on this parameterization and assuming sucrose is a good proxy for anthropogenic SOA, 70 % and 83 % of the mixing times within anthropogenic SOA in the PBL are < 1 h for January and July, respectively, when SOA

concentrations are significant (total organic aerosol concentration > 0.5 μg m$^{-3}$ at the surface). These percentages for anthropogenic SOA are likely lower limits since studies have shown that the Stokes-Einstein relation (which is used here to calculate diffusion coefficients of organic molecules from viscosities) can underpredict diffusion coefficients of organic molecules in sucrose-water mixtures by at least a factor of 10 to 100 at viscosities ≥ 10$^6$ Pa s (Chenyakin et al., 2017; Price et al., 2016).

To improve the predictions presented above the following are needed: 1) viscosities as a function of temperature and RH for α-pinene SOA and anthropogenic SOA generated using low mass concentrations and 2) studies that quantify the accuracy of the Stokes-Einstein equation for predicting diffusion coefficients in SOA. Studies that explore further the effect of oxidation level, oxidation type, and gas-phase precursor on viscosity and diffusion within biogenic and anthropogenic SOA would also be beneficial.


*Competing interests:* The authors declare that they have no conflict of interest.

**Acknowledgments.** This work was funded by the Natural Science and Engineering Research Council of Canada, DOE (ASR/BER) DE-SC0016559 & EPA STAR 83587701-0. This manuscript has not been reviewed by EPA and thus no endorsement should be inferred. Support from the MJ Murdock Charitable Trust (Grant 2012183) for computing infrastructure at Portland State University is acknowledged, as well as assistance from S.J. Hanna with the Amazon temperature and humidity data.

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

**Figures**

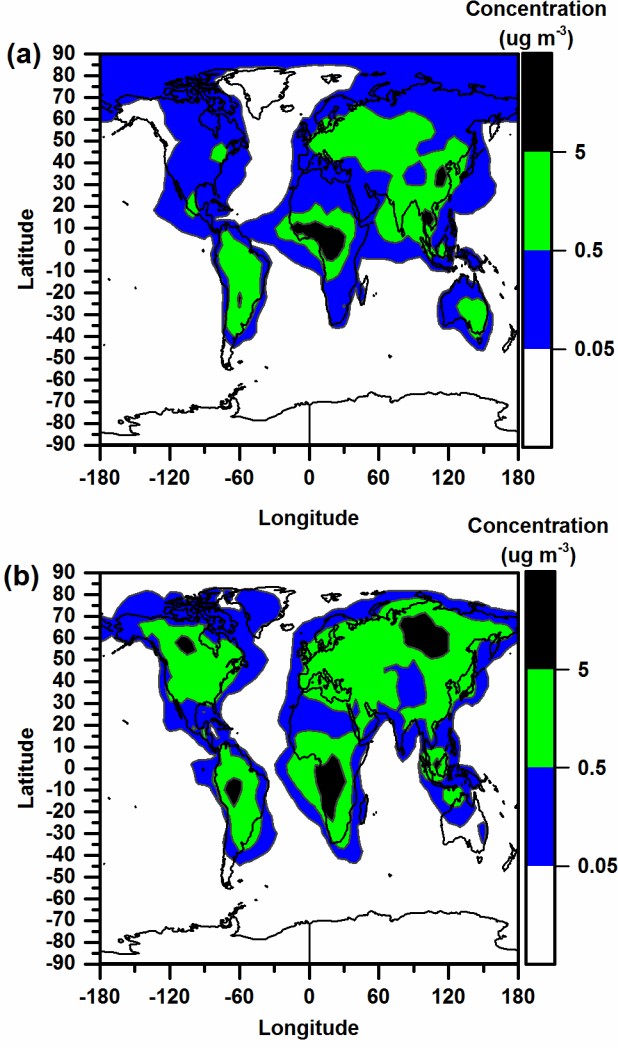

**Figure 1.** Monthly averaged total organic aerosol concentrations (color scale) at the Earth's surface in (a) January and (b) July, as calculated using GEOS-Chem.

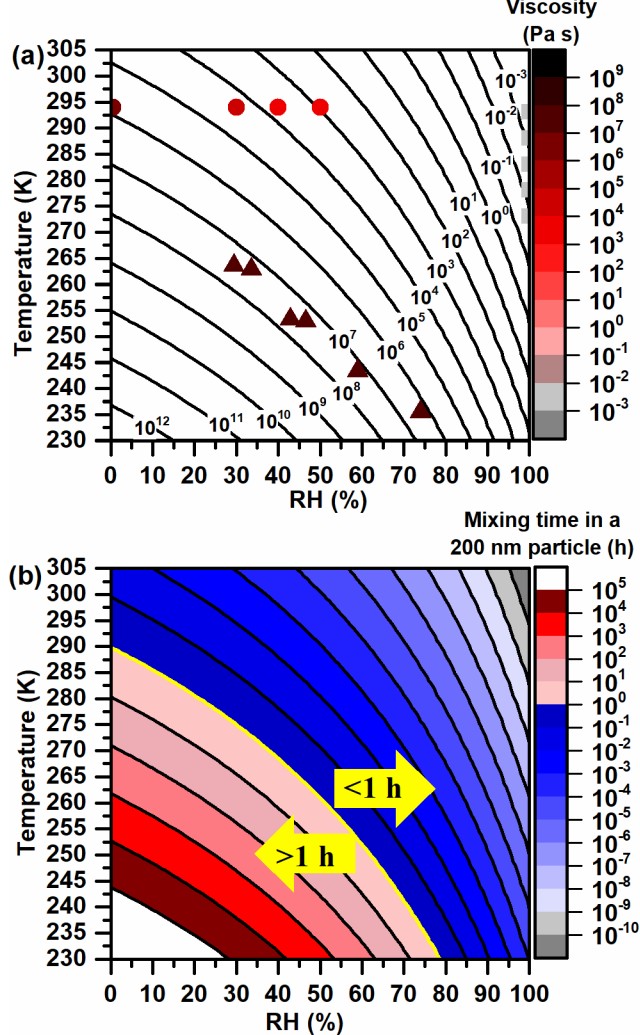

**Figure 2.** Plot of RH vs temperature with contour lines representing (a) our viscosity parameterization for α-pinene SOA particles and (b) mixing times calculated for organic molecules within 200 nm diameter α-pinene SOA particles. The symbols in (a) represent the laboratory data used to develop the parameterization: squares represent the water viscosities from Crittenden et al. (2012); triangles represent the viscosity data of α-pinene SOA from Järvinen et al. (2016) and the circles represent the viscosity data from Grayson et al. (2016). The viscosity parameterization is based on α-pinene SOA generated using mass concentrations of ~1000 μg m$^{-3}$.

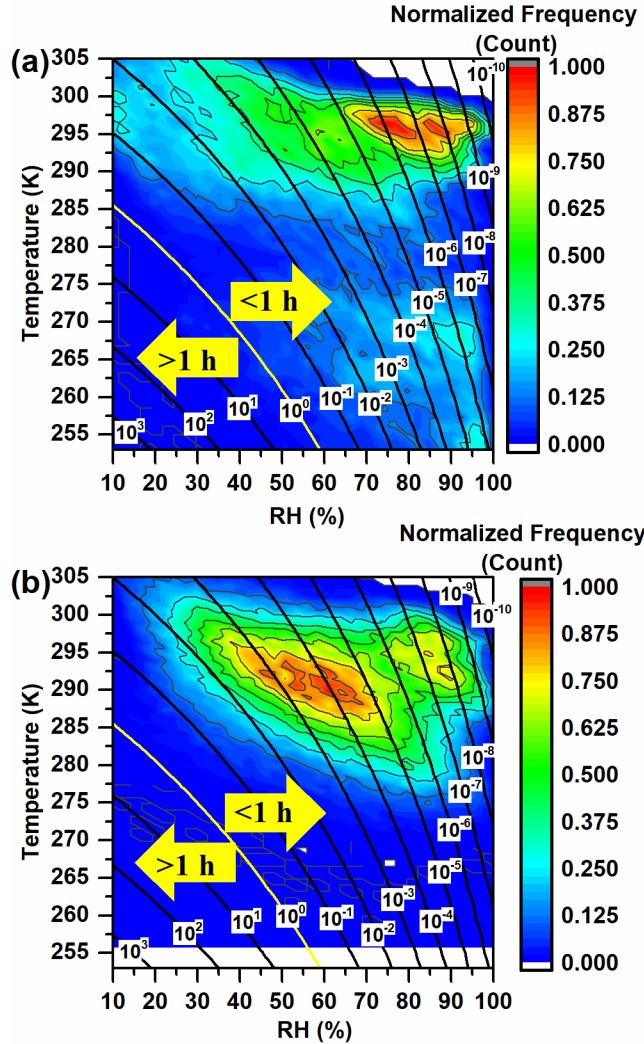

**Figure 3.** Six-hour normalized frequency counts of temperature and RH in the planetary boundary layer (PBL) (color scale) together with the mixing times for organic molecules within 200 nm α-pinene SOA particles (contours). Panel A shows the conditions for January and panel B shows the conditions for July. Mixing times (contours) are reported in hours. Frequency counts in the PBL were only included for the conditions where the mass concentration of total organic aerosol was > 0.5 μg m$^{-3}$ at the surface. The viscosity parameterization used to calculate mixing times was based on α-pinene SOA generated using mass concentrations of ~1000 μg m$^{-3}$.

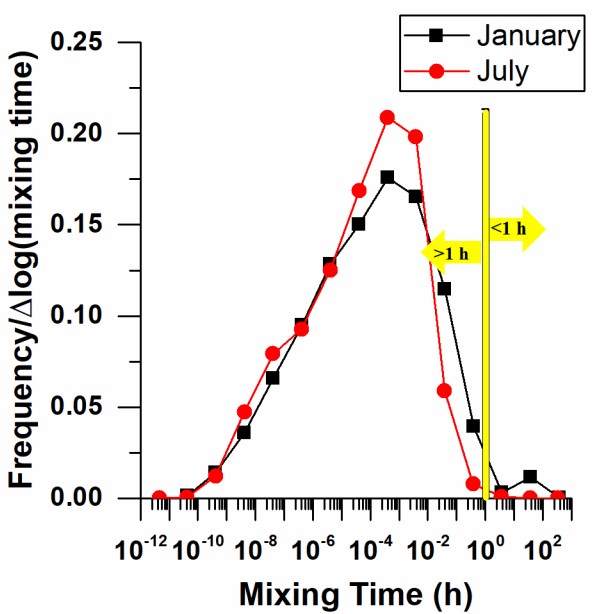

**Figure 4.** Normalized frequency distributions of mixing times within α-pinene SOA in the planetary boundary layer (PBL). Black symbols correspond to January and red symbols corresponds to July. Frequency counts in the PBL were only included for the conditions where the mass concentration of total organic aerosol was $> 0.5$ µg m$^{-3}$ at the surface. The viscosity parameterization used to calculate mixing times was based on α-pinene SOA generated using mass concentrations of $\sim$1000 µg m$^{-3}$.

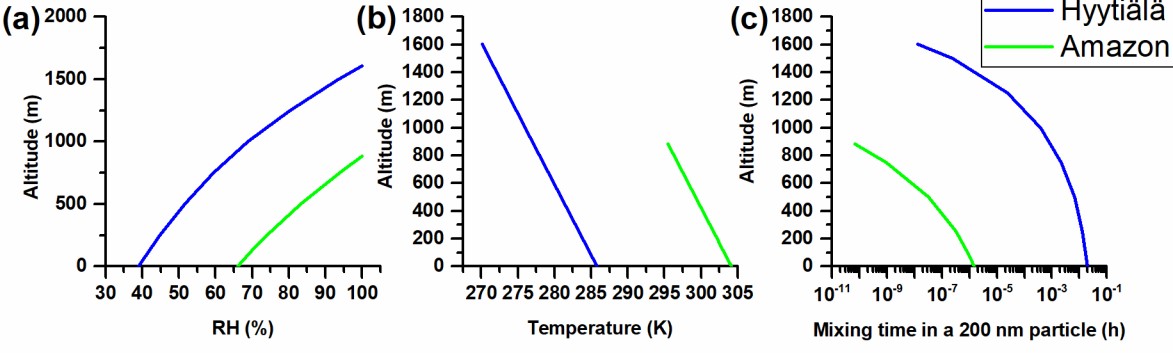

**Figure 5.** Temperature, RH and estimated mixing times for α-pinene SOA as a function of altitude for Hyytiälä (boreal forest) and the Amazon (rainforest). The temperature and RH at ground level are the average afternoon values in the driest month of the year for the respective locations. The vertical profiles of temperature and RH are plotted until the RH is 100 % for these locations. The height at which RH reaches 100 % is only slightly lower than the average height of the planetary boundary layer predicted by GEOS-5 meteorology data. For details see the Supporting Information, Section S1. The viscosity parameterization used to calculate mixing times was based on α-pinene SOA generated using mass concentrations of $\sim$1000 µg m$^{-3}$.

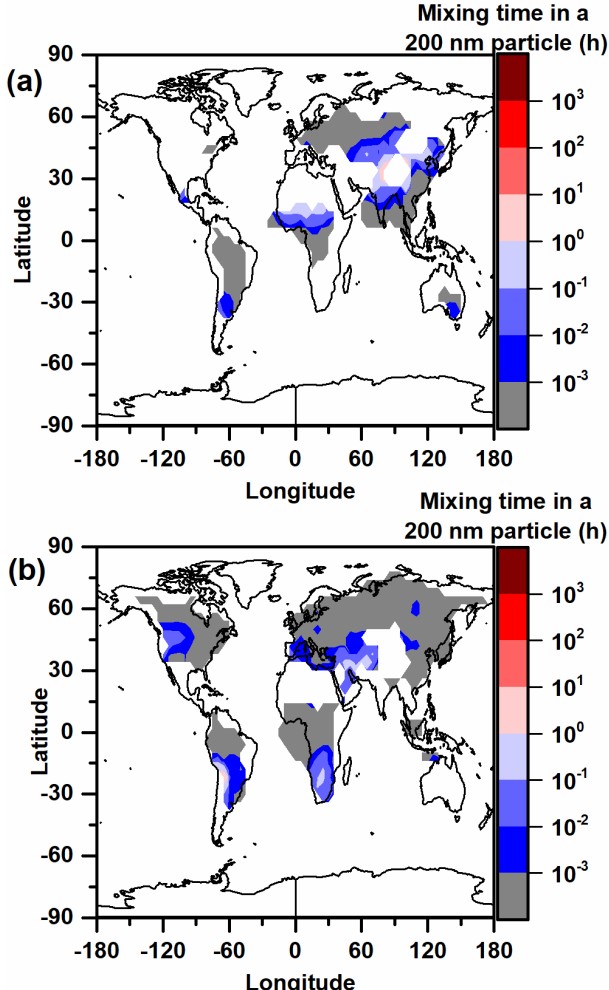

5    **Figure 6.** Mixing times of organic molecules within 200 nm α-pinene SOA particles at the top of the planetary boundary layer as a function of latitude and longitude in (a) January and (b) July. The color scale represents mixing times. Mixing times are only shown for locations with total organic aerosol concentrations > 0.5 ug m$^{-3}$ at the surface. The viscosity parameterization used to calculate mixing times were based α-pinene SOA generated using mass concentrations of ~1000 μg m$^{-3}$.

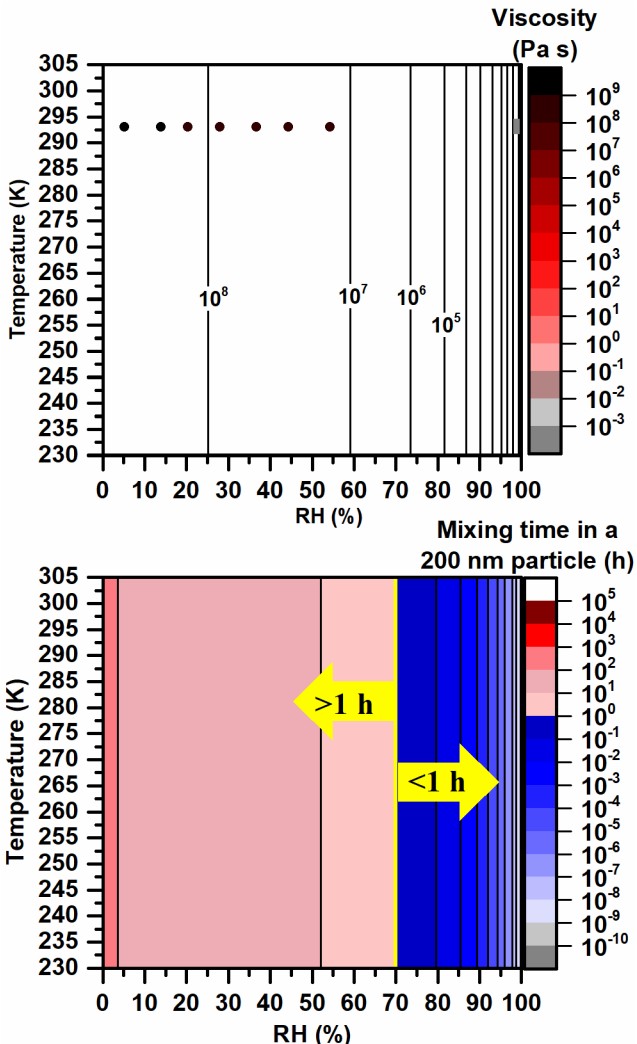

**Figure 7.** Plot of RH vs temperature with contour lines representing (a) the viscosity parameterization for α-pinene SOA particles based on the data from Zhang et al. (2015) and (b) mixing times calculated for organic molecules within 200 nm diameter α-pinene SOA particles. The symbols in (a) represent the laboratory data used to develop the parameterization: the square represents the water viscosity at room-temperature from Crittenden et al. (2012), and the circles represent the viscosity data from Zhang et al. (2015). The viscosity parameterization is based α-pinene SOA generated using mass concentrations of ~70 μg m⁻³.

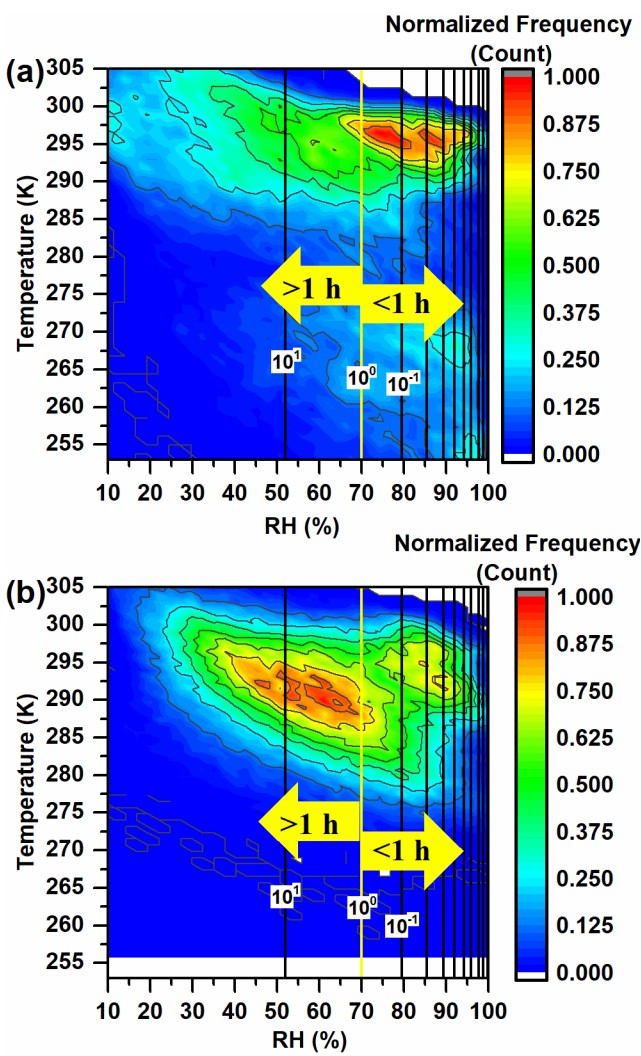

**Figure 8.** Six-hour normalized frequency counts of temperature and RH in the planetary boundary layer (PBL) (color scale) together with the mixing times for organic molecules within 200 nm α-pinene SOA particles (contours) calculated based on the parameterization generated using the viscosities from Zhang et al. (2015). Panel A shows the conditions for January and panel B shows the conditions for July. Mixing times (contours) are reported in hours. Frequency counts in the PBL were only included for the conditions where the mass concentration of total organic aerosol was > 0.5 μg m$^{-3}$ at the surface. The viscosity parameterization used to calculate mixing times was based on α-pinene SOA generated using mass concentrations of ~70 μg m$^{-3}$.

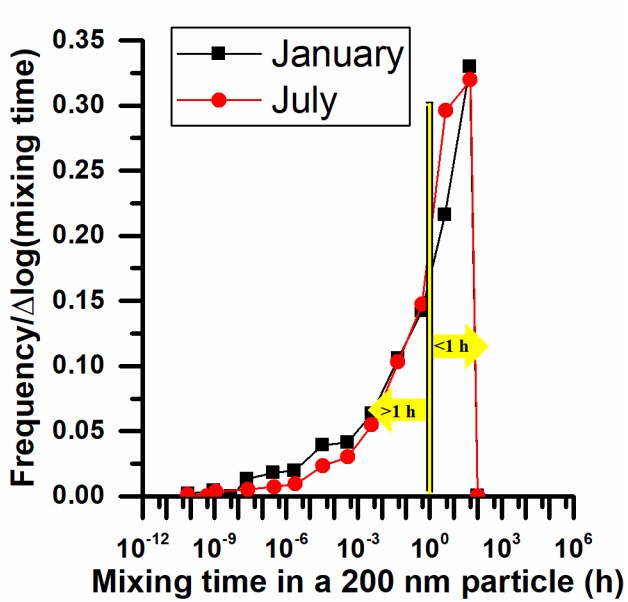

**Figure 9.** Normalized frequency distributions of mixing times within α-pinene SOA in the planetary boundary layer (PBL) for the parameterization generated using the upper limit of the viscosity data from Zhang et al. (2015). Black symbols correspond to January and red symbols corresponds to July. Frequency counts in the PBL were only included for the conditions where the mass concentration of total organic aerosol was > 0.5 μg m$^{-3}$ at the surface. The viscosity parameterization used to calculate mixing times was based on α-pinene SOA generated using mass concentrations of ~70 μg m$^{-3}$.