# Peer review of "Mixing times of organic molecules within secondary organic aerosol particles: a global planetary boundary layer perspective"

_Atmospheric Chemistry and Physics, 2017_

## Referee Comment (RC1) · Anonymous Referee #1 · 9 May 2017

Maclean et al. has estimated mixing times of organic molecules within secondary organic aerosol particles. In chemical transport models SOA particles are often assumed to be homogeneously well-mixed on the timescale of <1h, which could be in question if SOA particles adopt glassy or amorphous semisolid states. Combining laboratory data, meteorological conditions, and chemical transport modeling, this study predicted that mixing times should be indeed within <1h in the planetary boundary layer. They concluded that the assumption of well-mixed SOA in chemical transport models seems reasonable for biogenic SOA in most locations in the PBL. This is a very interesting study, the method seems reasonable, and the manuscript is clearly written and easy to follow. I have several comments as below, which should be implemented in the revised

manuscript before publication in ACP.

- The analysis is focused on 200nm-diameter particles and I agree that this may be most frequent size to be observed in ambient environments. Aged particles can have much larger diameters of up to 1 um, as observed for example in remote areas or Tokyo (see Fig. 7 in Takegawa et al., J. Geophys. Res., 111, D11206, 2006). Thus, I would suggest that the same analysis should be conducted with a larger diameter, say 500 nm-diameter particles. Then same figures of Fig. 3 could be presented and lines can be added in Fig. 4 (if the results are too similar with 200 nm, then they can be placed in the supplement/appendix). Mixing times should be larger for larger particles and I would be curious to know if mixing timescales would be still below 1 h. This should be easy and straightforward to do for authors and it will certainly strengthen their conclusion.

- It is very interesting to compare Fig. 6 in this study with Fig. 3d in Shiraiwa et al. (Nat. Communn., 8:15002, 2017). Shiraiwa et al. predicted the glass transition temperatures of SOA in a global model and estimated mixing timescales using annual average of RH and T for 2005-2009, while this study considers seasonal dependence, but did not simulate Tg or viscosity directly but viscosity was parameterized based on a-pinene viscosity measurements. I think there should be some discussion with a paragraph or two comparing these two studies. General trends seem to be consistent: longer timescales in west US, Sahara, and Mideast and shorter timescales in Europe and higher latitudes (Why there are no information over some places, such as Europe in panel a and over Amazon in both panels?). However, this study seems to estimate mixing timescales shorter in general. Please add some discussions.

- Abstract, L23: "SOA concentrations are significant." is ambiguous. I suggest being specific here (> 0.5 ug m-3).

- P2, L4: I suggest replacing "the lowest" to "low". Not only the lowest ones, but low and semivolatile products would also condense.

- P5, L3: "under predict" should be "underpredict".

- Figure 6 is not very easy to read and I feel this is because of overlapping yellow lines, arrows, and letters. Can you just remove these yellow things, and just put colors for places with SOA concentrations above 0.5 ug m-3? This would improve accessibility of this important figure.

- It may be good and helpful for readers to have a summary/conclusion section in the end of the manuscript.

- I suggest combining Section S1 with the main text, or include it as Appendix (particularly bring eq S1 and S2).

- I would suggest moving Fig. S3, S4 (also S5?) in the main text (maybe in Appendix?). There seem to be non-negligible cases with mixing timescales >1 h for anthropogenic SOA (given that sucrose is a good proxy for that).

---

## Referee Comment (RC2) · Anonymous Referee #2 · 30 May 2017

The authors report on mixing timescales within SOA particles using a parameterization that is developed based on literature data. They conclude that within the planetary boundary layer biogenic SOA particles can usually be considered well-mixed, having mixing timescales < 1h. Their work has potentially important implications for thinking about how air quality and climate models treat SOA formation and addresses an important topic. My major concerns relate to the robustness of the parameterization and how this might impact the conclusions here, especially in the context of (i) the exceptionally different, and still unexplained, viscosities between the Grayson et al. and Zhang et al. studies, the key ones for this work and (ii) the uncertainty within an individual study of SOA viscosity. I do not find that the current work sufficiently addresses the question of

[Figure]

robustness, even with the sensitivity test that is included. Associated, I have concerns that their statement that none of their conclusions are significantly impacted by data uncertainty is not sufficiently justified. Specific comments are below.

Fig. 1: Given that the parametrization depends on RH and T, it would be useful if Fig. 1 were augmented with additional panels showing the average PBL RH and T as a function of lat/lon.

P3/L19: Looking at Fig. 2, it is difficult to fully understand the parameterization that has been developed. It seems apparent that the viscosity of the a-pinene SOA measured at 293 K at a given RH differs dramatically between studies, with the reported values varying over orders of magnitude. (I'm comparing the "brown" circles to the more red "stars and pentagons.") In fact, the authors acknowledge this fact in section 3.4 ("Sensitivity analysis. . ."), and attempt to address it. However, I have substantial concerns, nonetheless. First, it is evident from Fig. 3 that the vast majority of the observations are in the T-range 290-300 K. This is the range of both the Grayson and Zhang observations. The Zhang et al. observations indicate that the viscosity at 293 K and 58% RH is 1 x 10ˆ7 Pa s, which translates to a mixing time of 5 h for a 200 nm diameter particle. A condition of 58% RH and T = 293 K is very close to the high probability region in Fig. 2B (July). Thus, it would seem that the probability of having mixing time scales >1 h in July (based on Zhang et al.) would be substantial, much more than indicated by the authors in Section 3.4. Most likely, this is because of the incorporation of the Jarvinen et al. low-T data, which appears to have a similar viscosity as the SOA from Zhang et al. at the same RH but a much lower temperature. Including the Jarvinen data, which is at temperatures well-below the most probable range, leads to the parameterized viscosity at this most probable (July) condition being underestimated relative to if only the Zhang et al. observations were used. (This is difficult to assess because the authors do not provide a Figure similar to Fig. 2 that shows the Grayson-excluded parameterization, nor do they provide their best fit parameters.) I suggest that the inclusion of histograms for the alternative (sensitivity) case, similar to Fig. 4,

is necessary. Additionally, I strongly suggest that a sensitivity case that excludes the pure water observations in developing the parametrization is needed. With this, the Grayson et al. and Zhang et al. results should be considered separately. This would require ignoring any T-dependence, but as most of the RH/T pairs overlap with these data sets, and the variability in RH is much greater than the variability in T, it would be a reasonable approximation. The authors must show the contours associated with their alternative parameterizations (as they do in Fig. 2 for their reference case).

Further, while I appreciate the sensitivity test that was done, it should be noted that the reported uncertainty in the Zhang et al. measurements is +/- 2 orders of magnitude. At the high end, this would imply that SOA in much of the atmosphere would not mix on a 1 h time scale. On the low end, nearly all SOA would always be well mixed. This is because a 1 h mixing time scale corresponds approximately to a viscosity of 2e6 Pa s, and thus variability around this value can have a large impact on the conclusions; the uncertainties on the Zhang et al. measurements overlap this critical value up to an RH of 58%.

Continuing with this, the results from Grayson et al. also suggest that the viscosity increases as the mass concentration decreases; this is offered as a potential (although not demonstrated) explanation for the substantially larger viscosities in Zhang et al. and in Renbaum-Wolff et al. The Zhang et al. measurements are still at SOA concentrations above ambient. Isn't it possible that the viscosity of SOA at ambient concentrations is even higher than that reported in Zhang et al.? Or, doesn't it suggest that the "sensitivity" case is actually the better base case, since the concentrations in Zhang et al. are closer to ambient than in Grayson et al.? Overall, I have substantial concerns that the authors are under-emphasizing the potential uncertainty in their estimates in a manner that may influence their conclusions. I think that these issues need to be explored further before this work should be published.

Fig. 2 and Eqn. 4: Regarding the translation between viscosity and mixing time scale, I have some concerns about the authors' illustration. Based on Fig. 2, a viscosity of

~2e7 Pa s corresponds to a mixing time scale of 1 h for a 200 nm particle. Using the stated hydrodynamic radius (0.38 nm), the calculated diffusion coefficient for viscosity = 2e7 Pa s is 2.8e-20 mˆ2/s and the mixing timescale for a 200 nm particle is 10 h. Thus, the yellow line in Fig. 2b seems to delineate between >10 h and <10 h, not >1 h and <1 h. My assessment seems consistent with the color scale in Fig. 2b. Similarly, the lines in Fig. 3a and 3b are incorrectly labeled: the line labeled >< 1 h is actually for 10 h. This should not materially affect any conclusions, but should be fixed.

The authors choose 0.5 micrograms/m3 as their dividing line between what to consider and what not to consider. While reasonable, this is nonetheless an arbitrary choice. Therefore, I suggest that it would be useful if the authors were to graph calculated viscosity vs. mass concentration. Is there any sort of trend that can be used to justify this dividing line?

Fig. 5: Do the authors not find it surprising that RH and T are not less variable with altitude within the PBL during the period shown (13:00-15:00 local time)? I typically think of the PBL as "well mixed" with respect to e.g. RH in the afternoon when mixing is vigorous. Is this a result of averaging over many months?

––––––––––––––––––––––––––––––

---

## Author Response (AR1)

Professor Nga Lee (Sally) Ng
Co-Editor of Atmospheric Chemistry and Physics

5  Dear Sally,

Listed below are our responses to the comments from the reviewers of our manuscript. We thank the reviewers for carefully reading our manuscript and for their very helpful suggestions! For clarity and visual distinction, the referee comments or questions are 10 listed here in black and are preceded by bracketed, italicized numbers (e.g. *[1]*). Authors' responses are in red below each referee statement with matching numbers (e.g. *[A1]*).

Sincerely,

15  Allan Bertram
Professor of Chemistry
University of British Columbia

Anonymous Referee #1

Maclean et al. has estimated mixing times of organic molecules within secondary organic aerosol particles. In chemical transport models SOA particles are often assumed to be 25 homogeneously well-mixed on the timescale of <1h, which could be in question if SOA particles adopt glassy or amorphous semisolid states. Combining laboratory data, meteorological conditions, and chemical transport modeling, this study predicted that mixing times should be indeed within <1h in the planetary boundary layer. They concluded that the assumption of well-mixed SOA in chemical transport models seems reasonable 30 for biogenic SOA in most locations in the PBL. This is a very interesting study, the method seems reasonable, and the manuscript is clearly written and easy to follow. I have several comments as below, which should be implemented in the revised manuscript before publication in ACP.

35  *[1]* The analysis is focused oan 200nm-diameter particles and I agree that this may be most frequent size to be observed in ambient environments. Aged particles can have much larger diameters of up to 1 um, as observed for example in remote areas or Tokyo (see Fig. 7 in Takegawa et al., J. Geophys. Res., 111, D11206, 2006). Thus, I would suggest that the same analysis should be conducted with a larger diameter, say 500 nm-40 diameter particles. Then same figures of Fig. 3 could be presented and lines can be added in Fig. 4 (if the results are too similar with 200 nm, then they can be placed in the supplement/appendix). Mixing times should be larger for larger particles and I would be curious to know if mixing timescales would be still below 1 h. This should be easy and straightforward to do for authors and it will certainly strengthen their conclusion.

[A1]  *To address the referee's comments we have calculated mixing times for 500 nm-diameter particles as suggested and added the results to the revised manuscript. See Section 3.4 and Figure S3 in the revised manuscript.*

*[2]* It is very interesting to compare Fig. 6 in this study with Fig. 3d in Shiraiwa et al. (Nat. Communn., 8:15002, 2017). Shiraiwa et al. predicted the glass transition temperatures of SOA in a global model and estimated mixing timescales using annual average of RH and T for 2005-2009, while this study considers seasonal dependence, but did not simulate Tg or viscosity directly but viscosity was parameterized based on a-pinene viscosity measurements. I think there should be some discussion with a paragraph or two comparing these two studies. General trends seem to be consistent: longer timescales in west US, Sahara, and Mideast and shorter timescales in Europe and higher latitudes (Why there are no information over some places, such as Europe in panel a and over Amazon in both panels?). However, this study seems to estimate mixing timescales shorter in general. Please add some discussions.

*[A2] In the revised manuscript, we have added a new section (Section 3.7) where we compared our studies with the studies by Shiraiwa et al. as suggested.*

*[3]* - Abstract, L23: "SOA concentrations are significant." is ambiguous. I suggest being specific here (> 0.5 ug m-3).

*[A3] This change has been made as suggested.*

*[4]* - P2, L4: I suggest replacing "the lowest" to "low". Not only the lowest ones, but low and semivolatile products would also condense.

*[A4] This change has been made as suggested.*

*[5]* - P5, L3: "under predict" should be "underpredict".

*[A5] This change has been made as suggested.*

*[6]* - Figure 6 is not very easy to read and I feel this is because of overlapping yellow lines, arrows, and letters. Can you just remove these yellow things, and just put colors for places with SOA concentrations above 0.5 ug m-3? This would improve accessibility of this important figure.

*[A6] This change has been made as suggested.*

[7] - It may be good and helpful for readers to have a summary/conclusion section in the end of the manuscript.

*[A7] A summary/conclusion section has been added as suggested.  See Section 4.0 in the revised manuscript.*

*[8]* - I suggest combining Section S1 with the main text, or include it as Appendix (particularly bring eq S1 and S2).

*[A8] This change has been made.*

*[9]* - I would suggest moving Fig. S3, S4 (also S5?) in the main text (maybe in Appendix?). There seem to be non-negligible cases with mixing timescales >1 h for anthropogenic SOA (given that sucrose is a good proxy for that).

*[A9] If possible, we would prefer to keep these figures in the Supplement to avoid making the main document too long. However, we can move these figures to the main text if the Editor prefers.*

Anonymous Referee #2

*[10]* The authors report on mixing timescales within SOA particles using a parameterization that is developed based on literature data. They conclude that within the

20 planetary boundary layer biogenic SOA particles can usually be considered well-mixed, having mixing timescales < 1h. Their work has potentially important implications for thinking about how air quality and climate models treat SOA formation and addresses an important topic. My major concerns relate to the robustness of the parameterization and how this might impact the conclusions here, especially in the context of (i) the

25 exceptionally different, and still unexplained, viscosities between the Grayson et al. and Zhang et al. studies, the key ones for this work and (ii) the uncertainty within an individual study of SOA viscosity. I do not find that the current work sufficiently addresses the question of robustness, even with the sensitivity test that is included. Associated, I have concerns that their statement that none of their conclusions are significantly impacted by

30 data uncertainty is not sufficiently justified. Specific comments are below.

*[A10] Thank you for raising these important and excellent questions/comments. We have addressed these questions/comments below.*

35 *[11]* Fig. 1: Given that the parametrization depends on RH and T, it would be useful if Fig. 1 were augmented with additional panels showing the average PBL RH and T as a function of lat/lon.

*[A11] As suggested, we have added figures to the revised manuscript (Figures S1 and*
40 *S2) that show global maps of the average RH and T for January and July at the Earth's surface and the top of the planetary boundary layer.*

*[12]* P3/L19: Looking at Fig. 2, it is difficult to fully understand the parameterization that has been developed. It seems apparent that the viscosity of the a-pinene SOA measured
45 at 293 K at a given RH differs dramatically between studies, with the reported values varying over orders of magnitude. (I'm comparing the "brown" circles to the more red

"stars and pentagons.") In fact, the authors acknowledge this fact in section 3.4 ("Sensitivity analysis..."), and attempt to address it. However, I have substantial concerns, nonetheless. First, it is evident from Fig. 3 that the vast majority of the observations are in the T-range 290-300 K. This is the range of both the Grayson and
5 Zhang observations. The Zhang et al. observations indicate that the viscosity at 293 K and 58% RH is 1 x 10ˆ7 Pa s, which translates to a mixing time of 5 h for a 200 nm diameter particle. A condition of 58% RH and T = 293 K is very close to the high probability region in Fig. 2B (July). Thus, it would seem that the probability of having mixing time scales >1 h in July (based on Zhang et al.) would be substantial, much more than
10 indicated by the authors in Section 3.4. Most likely, this is because of the incorporation of the Jarvinen et al. low-T data, which appears to have a similar viscosity as the SOA from Zhang et al. at the same RH but a much lower temperature. Including the Jarvinen data, which is at temperatures well-below the most probable range, leads to the parameterized viscosity at this most probable (July) condition being underestimated relative to if only the
15 Zhang et al. observations were used. (This is difficult to assess because the authors do not provide a Figure similar to Fig. 2 that shows the Grayson-excluded parameterization, nor do they provide their best fit parameters.) I suggest that the inclusion of histograms for the alternative (sensitivity) case, similar to Fig. 4, is necessary. Additionally, I strongly suggest that a sensitivity case that excludes the pure water observations in developing
20 the parametrization is needed. With this, the Grayson et al. and Zhang et al. results should be considered separately. This would require ignoring any T-dependence, but as most of the RH/T pairs overlap with these data sets, and the variability in RH is much greater than the variability in T, it would be a reasonable approximation. The authors must show the contours associated with their alternative parameterizations (as they do in Fig. 2 for their
25 reference case).

*[A12] To address the referee's concerns, in the revised manuscript, we have first focused on a parameterization that just includes the room-temperature and low-temperature viscosity data from Grayson et al. and Jarvinen et al., which corresponds to SOA*
30 *generated at high mass concentrations.  See Sections 3.1-3.4 in the revised manuscript. Then, we focused on a parameterization that just includes the SOA room-temperature viscosity data from Zhang et al., which corresponds to SOA generated under low mass concentrations.  See Section 3.5 in the revised manuscript.*

35 *[13]* Further, while I appreciate the sensitivity test that was done, it should be noted that the reported uncertainty in the Zhang et al. measurements is +/- 2 orders of magnitude. At the high end, this would imply that SOA in much of the atmosphere would not mix on a 1 h time scale. On the low end, nearly all SOA would always be well mixed. This is because a 1 h mixing time scale corresponds approximately to a viscosity of 2e6 Pa s,
40 and thus variability around this value can have a large impact on the conclusions; the uncertainties on the Zhang et al. measurements overlap this critical value up to an RH of 58%.

*[A13] In the revised manuscript, uncertainties in the viscosity data have been considered*
45 *in the sensitivity analysis. See Section 3.4 and Figure S4 in the revised manuscript. The*

*sensitivity analysis was performed using the upper viscosity limits of the Grayson et al. data and the upper RH limits of the Järvinen et al. data.*

*[14]* Continuing with this, the results from Grayson et al. also suggest that the viscosity increases as the mass concentration decreases; this is offered as a potential (although not demonstrated) explanation for the substantially larger viscosities in Zhang et al. and in Renbaum-Wolff et al. The Zhang et al. measurements are still at SOA concentrations above ambient. Isn't it possible that the viscosity of SOA at ambient concentrations is even higher than that reported in Zhang et al.? Or, doesn't it suggest that the "sensitivity" case is actually the better base case, since the concentrations in Zhang et al. are closer to ambient than in Grayson et al.? Overall, I have substantial concerns that the authors are under-emphasizing the potential uncertainty in their estimates in a manner that may influence their conclusions. I think that these issues need to be explored further before this work should be published.

*[A14] To address the referee's comments we have added a section to the revised manuscript that discusses the effect of mass concentration used to generate the SOA on viscosity. See Section 3.5 in the revised manuscript.*

*[15]* Fig. 2 and Eqn. 4: Regarding the translation between viscosity and mixing time scale, I have some concerns about the authors' illustration. Based on Fig. 2, a viscosity of ~ 2e7 Pa s corresponds to a mixing time scale of 1 h for a 200 nm particle. Using the stated hydrodynamic radius (0.38 nm), the calculated diffusion coefficient for viscosity = 2e7 Pa s is 2.8e-20 m^2/s and the mixing timescale for a 200 nm particle is 10 h. Thus, the yellow line in Fig. 2b seems to delineate between >10 h and <10 h, not >1 h and <1 h. My assessment seems consistent with the color scale in Fig. 2b. Similarly, the lines in Fig. 3a and 3b are incorrectly labeled: the line labeled >< 1 h is actually for 10 h. This should not materially affect any conclusions, but should be fixed.

*[A15] Yes, this was a mistake. The mistake has been fixed in the revised manuscript.*

*[16]* The authors choose 0.5 micrograms/m3 as their dividing line between what to consider and what not to consider. While reasonable, this is nonetheless an arbitrary choice. Therefore, I suggest that it would be useful if the authors were to graph calculated viscosity vs. mass concentration. Is there any sort of trend that can be used to justify this dividing line?

*[A16] We chose a mass concentration of 0.5 µg m$^{-3}$ for filtering because the mass concentration of organic aerosol at the surface was > 0.5 µg m$^{-3}$ in all but one of the previous field measurements of organic aerosol at remote locations (Spracklen et al. 2011). To address the referee's comment this information has been added to the revised manuscript. Specifically, we have added the following text to Section 3.2:*
"We chose a mass concentration of > 0.5 µg m$^{-3}$ for filtering because the mass concentration of total organic aerosol at the surface was > 0.5 µg m$^{-3}$ in all but one of the previous field measurements of organic aerosol at remote locations (Spracklen et al., 2011)."

*[17]* Fig. 5: Do the authors not find it surprising that RH and T are not less variable with altitude within the PBL during the period shown (13:00-15:00 local time)? I typically think of the PBL as "well mixed" with respect to e.g. RH in the afternoon when mixing is vigorous. Is this a result of averaging over many months.

*[A17] Fig. 5 was calculated using a dry adiabatic lapse rate and assuming the mixing ratio of water is independent of height in the PBL. In the revised manuscript we have tried to clarified this point in Section S1. Below is the revised relevant text from Section S1.*
"The vertical profiles of RH were calculated using the average afternoon surface RHs mentioned above, the vertical profiles of temperature (calculated with the dry adiabatic lapse rate), and assuming the mixing ratio of water is independent of height in the PBL. For the calculations of RH as a function of altitude, the water vapor pressure and water saturated vapour pressure were needed as a function of altitude. The water vapor pressure as a function of altitude was determined by multiplying the mixing ratio of water by the atmospheric pressure, 
[revised manuscript text omitted]
 n α-pinene and isoprene SOA for ambient temperatures and relative humiditesRHs in the for ambient conditions in the PBL. In addition, consider the effect of mass loading on the mixing times and develop a mixing time parameterization for low mass loadings. Mixing times within anthropogenic SOA and the effect of SOA mass concentration on mixing times are also discussed. Our study is complementary to the recent study by Shiraiwa et al. (2017) on the global distribution of particle phase state in atmospheric SOA, although our study focuses on mixing times within SOA in the PBL and uses a different approach to determine physicochemical properties of SOA.

**2. Materials and Methods**

**2.1 Parameterization for the viscosity of α-pinene SOA as a function of temperature and RH.**

The following data was used to develop a parameterization of the viscosity α-pinene SOA as a function of temperature and RH: a) room-temperature measurements of viscosity of SOA derived from α-pinene ozonolysis by Grayson et

5 al. (Grayson et al., 2016) (Table S1), b) low-temperature measurements of viscosity for SOA derived from α-pinene ozonolysis by Järvinen et al. (Järvinen et al., 2016) (Table S2), and c) temperature dependent measurements of viscosity for water from Crittenden et al. (Crittenden et al., 2012) (Table S3). Järvinen et al. (2016) measured the temperature and RH values at which α-pinene SOA has a viscosity of approximately $10^7$ Pa s. In these experiments, SOA was generated with a mass concentration of 707-1414 µg m$^{-3}$. Grayson et al. (2016) measured viscosity of α-

10 pinene SOA as a function of RH at 295 K. In these experiments, the SOA was generated with mass concentrations of 121 µg m$^{-3}$ and 520 µg m$^{-3}$. We use the viscosity measurements from Grayson et al. (2016) determined with a mass concentration of 520 µg m$^{-3}$ to be more consistent with the mass concentrations used by Järvinen et al. (2016). Although there are other room-temperature measurements of the viscosity of α-pinene SOA (Bateman et al., 2015; Hosny et al., 2016; Kidd et al., 2014; Pajunoja et al., 2014; Renbaum-Wolff et al., 2013), we used the room-

15 temperature measurements from Grayson et al. (2016) because 1) viscosity was measured over a range of relative humidities in this study, 2) the mass concentrations used by Grayson et al. (2016) to generate the SOA were similar to the mass concentrations used by Järvinen et al. (2016), and 3) Grayson et al. (2016) measured the viscosity of the total SOA (both the water soluble component and the water insoluble component).

20 Due to the experimental conditions used by Grayson et al. (2016) and Järvinen et al. (2016), the parameterization developed here is applicable to SOA generated using a mass concentration of ~ 1000 µg m$^{-3}$. We focused on ~ 1000 µg m$^{-3}$ because both low-temperature and room-temperature viscosity measurements have been carried out using this mass concentration. The effect of mass concentration on the viscosity α-pinene SOA is discussed in Section 3.5. (Bateman et al., 2015; Hosny et al., 2016; Kidd et al., 2014; Pajunoja et al., 2014; Renbaum-Wolff et al., 2013)

(Bateman et al., 2015) (Grayson et al., 2016) (Zhang et al., 2015)RH (Bateman et al., 2015) (Grayson et al., 2016) (Zhang et al., 2015)80-220 µg m$^{-3}$520 µg m$^{-3}$70 µg m$^{-3}$(Järvinen et al., 2016) 700-1400 µg m$^{-3}$) Grayson et al. (Grayson et al., 2016) measured viscosities of α-pinene SOA at two different mass loadings and found the SOA generated at the lower mass loading had a higher viscosity. (Grayson et al., 2016)520 µg m$^{-3}$Järvinen (2016)µ (2016)520 µg m$^{-3}$ by

[revised manuscript text omitted]
 for January and July. The conditions for the top of the planetary boundary layer can be found in Fig. S1.

**3. Results and Discussion**

**3.1 Parameterization of viscosity and mixing times within α-pinene SOA particles as a function of RH and temperature**

Shown in Fig. 2a (contours) is the RH and temperature dependent parameterization for α-pinene SOA viscosities

5    based on  viscosities measured at roomtemperature (Grayson et al., 2016)and lowtemperature (Järvinen et al., 2016), as well as the viscosity of water as a function of temperature (Crittenden et al., 2012). From the viscosity parameterization, the diffusion coefficients of organic molecules within α-pinene SOA particles were calculated using the Stokes-Einstein equation:

$$D = \frac{kT}{6\pi\eta R_H}$$

10        (3)

where $D$ is the diffusion coefficient, $k$ is the Boltzmann constant, $T$ is temperature in Kelvin, $\eta$ is the dynamic viscosity and $R_H$ is the hydrodynamic radius of the diffusing species. For the calculations, a hydrodynamic radius of 0.38 nm was used for the diffusing organic molecules within SOA, based on an assumed molecular weight of 175 g mol[-1] (Huff Hartz et al., 2005), a density of 1.3 g cm[-3] (Chen and Hopke, 2009; Saathoff et al., 2009) and spherical symmetry. The

15    Stokes-Einstein equation should give reasonable values when the radius of the diffusing molecules is roughly greater than or equal to  the radius of the  matrix molecules and when the viscosity of the matrix is relatively small ($\lesssim$ 400 Pa s) (Chenyakin et al., 2017; Price et al., 2016). When the viscosity of the matrix is large ($\gtrsim$ 10[6] Pa s), the Stokes-Einstein equation can underpredict diffusion coefficients of organic molecules in organic matrices (Champion et al., 1997; Chenyakin et al., 2017; Price et al., 2016). Hence, the diffusion coefficients and

20    mixing times estimated here should be considered lower and upper limits, respectively.

From the diffusion coefficients, the mixing times of organic molecules within an α-pinene SOA particle were calculated with the following equation (Shiraiwa et al., 2011):

$$\tau_{mix} = \frac{d^2}{4\pi^2 D}$$

        (4)

25    where $\tau_{mix}$ is the mixing time, d is the diameter of an SOA particle, and D is the diffusion coefficient estimated from Eq. (3). For these calculations, it was assumed that the α-pinene SOA particles have a diameter of 200 nm, which is roughly the median diameter in the volume distribution of ambient SOA-containing particles (Martin et al., 2010; Pöschl et al., 2010; Riipinen et al., 2011). Once the mixing time has elapsed, the concentration of the diffusing molecules at the centre of the particle is within 1/e of the equilibrium concentration (Shiraiwa et al., 2011). The

30    calculated mixing times (Fig. 2b) illustrate that, as expected,  inverse relationships exist between both mixing time and RH, as well as mixing time and temperature.

**3.2 RH and temperature in the PBL**

Shown in Fig. 3a and 3b are the normalized frequency counts of temperature and RH in the PBL for the months of January and July, 2006, respectively, based on the archive of meteorological fields (GEOS-5) used in the global

35    chemical transport model, GEOS-Chem, v10-01. We only included grid points in our analysis if the grid points were

within the PBL and the monthly average mass concentration of total organic aerosol was > 0.5 μg m$^{-3}$ at the surface, based on GEOS-Chem, v10-01 (Fig. 1). In other words, we included all the grid points in a column up to the top of the PBL when determining frequency counts if the monthly averaged total organic aerosol concentration was > 0.5 μg m$^{-3}$ at the surface. This filtering removes cases where SOA concentrations are not expected to be of major importance

5  for climate, health or visibility. We chose a mass concentration of > 0.5 μg m$^{-3}$ for filtering because the mass concentration of total organic aerosol at the surface was > 0.5 μg m$^{-3}$ in all but one of the previous field measurements of organic aerosol at remote locations (Spracklen et al., 2011).

This concentration was chosen as it had been determined by Spracklen et al. (Spracklen et al., 2011) that the majority

10  of surface measurements of organic aerosols, including in remote locations, had concentrations >0.5 μg m$^{-3}$. The normalized frequency counts illustrate that the temperature and RH in the PBL areis often in the range of 290-300 K and > 50 % RH for the month of January (Fig. 33a) and in the range of 285-300 K and > 30 % RH for the month of July (Fig. 33b). For reference, sShown in Figs.ure S1S1 and S2 are the average temperature and RH conditions at the Earth's surface and top of the planetary boundary layer, respectively, for January and July, based on the archive of

15  meteorological fields for 2006 (GEOS-5).

**3.3 Mixing times of organic molecules within α-pinene SOA particles in tthe PBL**

Also shown in Figs. 33a and 33b are the contour mixing times within 200 nm α-pinene SOA particles predicted with our lines produced using our parameterization (contours) of mixing times of organics within 200 nm α pinene SOA particles. These results, together with the frequency counts of temperature and RH throughout the vertical column of

20  the PBL, indicate that the mixing times of organic molecules within α-pinene SOA areis often < 1x10$^{-1}$ h for conditions in the PBL.

Shown in Fig. 44 are the normalized frequency distributions of mixing times within α-pinene SOA for January and July, based on the data in Figs. 33a and 33b. Figure 4 4 suggests that the mixing times within α-pinene SOA areis < 1 h for 94 98.5 % and 99.9 % of the occurrences in the PBL during January and July, respectively, when monthly

25  average total organic aerosol concentrations arewere > 0.5 μg m$^{-3}$ at the surface. Takegawa et al. (Takegawa et al., 2006) found that aged aerosol particles can have diameters larger than 200 nm, so the mixing time calculations were repeated for a particle size of 500 nm (Figure S2). It was found that the mixing times with the 500 nm particles was <1 h for 95.9% and 99.4% of the occurrences in the PBL during January and July, respectively.

Within the PBL, RH increases and temperature decreases with altitude, with both changes being substantial and

30  impacting mixing times in opposite directions. Shown in Fig. 55a-c are calculated monthly average afternoon (13:00-15:00, local time) vertical profiles of temperature, RH, and mixing times within α-pinene SOA over Hyytiälä (boreal forest), and the Amazon (rainforest) for the driest month of the year at eachthese locations (the method used to calculate vertical profiles is described in the Supporting Information, Section S2S1). Afternoon vertical profiles were chosen since this is the time of the day when RH is typically lowest and thus mixing times are the longest. Figure 5c

35  5c shows that mixing times within α-pinene SOA decrease significantly with altitude for these two locations. This is because the plasticizing effect of water on viscosity dominates the temperature effect for these conditions.

Shown in Fig. 66 are global maps of the monthly averaged mixing times of organic molecules within α-pinene SOA for conditions at the top of the PBL for the months of January and July. Figure 6 6 shows that 83 91.2 % and 92 97.5 % of the locations for January and July, respectively, have a mixing time < 0.1 h for conditions at the top of the PBL when monthly averaged total organic aerosol surface concentrations are > 0.5 µg m$^{-3}$. Within the PBL, vertical mixing of air masses occurs on the order of 30 min. Since the mixing times within α-pinene SOA particles for conditions at the top of the PBL are < 0.1 h for most locations where the SOA concentrations are significant (total organic aerosol concentration > 0.5 µg m$^{-3}$ at the surface), a reasonable upper limit to the mixing time within the α-pinene SOA studied here for most locations in the PBL is 30 min. During this 30 min interval, mixing times within α-pinene SOA particles can cycle between short and long values, though rarely being > 1 h (Figs. 33 and 44).

Shiraiwa et al (Shiraiwa et al., 2017) performed a similar study to investigate the mixing time in SOA particles in the atmosphere. The researchers used the relationship between volatility, molar mass and O:C ratio to predict the glass transition temperature of the SOA. The ratio of the glass transition temperature and the ambient temperature were related to the phase state of the SOA particle and from their mixing times were inferred. The mixing times were determined at several different pressures to study the mixing time at different altitudes. The two studies agree that mixing times are expected to be short in regions such as the Amazon, Europe and at high latitudes and longer mixing times will occur in the western United States as well as the Sahara desert. However, in general, this study predicts shorter mixing times overall than those described in Shiraiwa et al. (Shiraiwa et al., 2017), however this study focuses on one particular type of SOA whereas the study by Shiraiwa et al. (2017) does not focus on one type of SOA.

3.4 Sensitivity analysis for α-pinene SOA particles in the PBL

**3.4 Sensitivity analysis**

To calculate the mixing times discussed above, we assumed that the α-pinene SOA particles have a diameter of 200 nm. We also repeated these calculations assuming a diameter of 500 nm, since aged organic aerosol can have larger diameters , however, a (Takegawa et al., 2006). Based on the viscosity parameterization shown in Fig. 2a, mixing times within 500 nm α-pinene SOA particles are < 1 h for 95.9 % and 99.4 % of the occurrences in the PBL during January and July, respectively (Fig.ure S23).

The parameterization for of the viscosity usedof α pinene SOA was above was developed using based on viscosity measurements from by Bateman et al. (2015), Grayson et al. (2016), Zhang et al. (2015), Järvinen et al. (2016) and Crittenden et al. (2012). As a sensitivity analysis, we developed a second parameterization, using the same procedure as describe above, but using the upper limits to the viscosities reported by Grayson et al. (2016) and the upper limits to the RH ranges reported by Järvinen et al. (2016). This should result in an upper limit to the viscosity parametertization discussed above. The (Grayson et al., 2016) (Järvinen et al., 2016) uncertainties in the measurements by Crittenden et al. (Crittenden et al., 2012) were not considered since they are small compared to the uncertainties reported by Grayson et al. (2016) and Järvinen et al. (2016). . The viscosity measurements reported by Grayson et al. (Grayson et al., 2016) covered roughly 1 2 orders of magnitude at each RH, as well the values reported by Järvinen et al. (Järvinen et al., 2016) at each temperature covered a RH range of 5 10%. As a sensitivity analysis, we developed two additional parameterizations, using the same procedure as above. The first parameterization used

the lower viscosity values reported by Grayson et al. (Grayson et al., 2016) and the lower RH reported by Järvinen et al. (Järvinen et al., 2016), and the second parameterization used the upper viscosity and RH limits of Grayson et al. (Grayson et al., 2016) and upperJärvinen et al. (Järvinen et al., 2016), respectively. The frequency of the mixing times for January and July for both parameterizations are shown in Figures S3 and S4, respectively. For the upper limitsBased on, this second parameteriation, mixing times are < 1 h for 96.6 % and 99.5 % of the occurrences in the PBL during January and July, respectively, when the total organic aerosol was > 0.5 µg m$^{-3}$ at the surface (Fig.ure S34). for January, 96.6% of the conditions in the PBL resuted in mixing times <1 hr (previously 98.5%) and for July, 99.5% of the conditions resulted in mixing times <1 hr (previously 99.9%) when the total organic aerosol was >0.5 µg m$^{-3}$ at the surface. None of these results impact the overall conclusiongs of the paperThe measurements by Zhang et al. (2015) and Grayson et al. (2016) were both carried out at room temperature and over a similar range in RH. The viscosities reported by Zhang et al. (2015) were higher than the viscosities reported by Grayson et al. (2016) (see Table S1). As a sensitivity analysis, we developed a second parameterization, using the same procedure as describe above, but excluding the data from Grayson et al. (2016) in the fitting procedure. Based on this new parameterization, for January, 86 % of the conditions in the PBL resulting in mixing times <1 h (previously 93 %) and for July, 96 % of the conditions producing mixing times <1 h (previously 98 %) when the total organic aerosol was > 0.5 µg m$^{-3}$ at the surface. Using this new parameterization, we also found that the number of locations with mixing times <0.1 h decreased from 83 to 80 % and 92 to 89 % for January and July, respectively. None of these results significantly impact the overall conclusions of the paper.

**3.5 Effect of mass concentration used to generate the SOA**

The parameterizations developed above were based on SOA generated using a mass concentration of ~ 1000 µg m$^{-3}$. As mentioned, we focused on ~ 1000 µg m$^{-3}$ because low-temperature and room-temperature viscosity measurements have been carried out using this mass concentration. However, the viscosity of some types of SOA may depend on the mass concentration used to generate the SOA. For example, , results from Grayson et al. (Grayson et al., 2016) showed that under dry conditions, the viscosity of α-pinene SOA may increase by a factor of 5 as the production mass concentration decreased from 1200 µg m$^{-3}$ to 120 µg m$^{-3}$. increasedecrease 104In addition, mass concentrations of biogenic SOA are typically ≤ 10 µg m$^{-3}$ in the atmosphere (Spracklen et al., 2011). indicate that SOA generated at low mass loading has higher viscosities than SOA generated at high mass loadings.t7001400 µg m$^{-3}$As a starting point to quantify how often mixing times of organic molecules are < 1 h within α-pinene SOA generated using low mass concentrations, we developed a temperature-independent parameterization using the upper limit of theroom-temperature viscosity data for α-pinene SOA from Zhang et al. (Zhang et al., 2015) (Table SX5) and room-temperature viscosity data for water from (Crittenden et al., 2012) (Table SY3). Zhang et al. (2015) measured the viscosity of α-pinene SOA over a range of relative humidities (0-60 %), and the SOA used in these experiments was generated in the laboratory using a mass concentration of ~70 µg m$^{-3}$. 060%The median room-temperature viscosities reported by Zhang et al. are higher than the median room-temperature viscosities reported by Grayson et al. (2016) using a mass concentration of 520 µg m$^{-3}$ (Fig.ure SX5). Although not proven, a reasonable explanation for the difference in median viscosities is the difference in mass concentrations used to generate the SOA.

70 µg m⁻³ the (Crittenden et al., 2012)A temperature-independent parameterization was generated by fitting Eq. (1) to the upper limit of theroom-temperature viscosity data from Zhang et al. (Zhang et al., 2015) and water viscosity fromCrittenden et al. (Crittenden et al., 2012), 20121, but with the temperature (T) in Eq. (1) replaced by 293 K. The values for the parameters retrieved by fitting the modified Eq. (1) to the viscosity data are reported in Table SX6. The

5 temperature-independent parameterization generated using this method is shown in Fig.ure XS5Z7a-. Shown in Figs.ures 7b, 8a, and 8bF8F8 (contours) is the parameterization for are the viscosity and mixing times within 200 nm α-pinene SOA based on this temperature-independent viscosity parameterization. contours generated for this parameterization Also included in Figs. F8a and F8b are the normalized frequency counts of temperature and RH in the PBL for the months of January and July, 2006, respectively, when the monthly average mass concentration of total

10 organic aerosol was > 0.5 µg m⁻³ at the surface. Shown in Fig. G9 are the normalized frequency distributions of mixing times within α-pinene SOA for January and July, based on the data in Figs. F8a and F8b. Figure G9 suggests that the mixing times within α-pinene SOA is < 1 h for X45 and Y38 % of the occurrences in the PBL during January and July, respectively, when monthly average total organic aerosol concentrations were > 0.5 µg m⁻³ at the surface. The frequency of the different conditions and the expected mixing times and the frequency of the different mixing times

15 can be seen in Figures S6 and S7, respectively..41 and 34%However, several <caveats need to be emphasized: 1) thethe 
[revised manuscript text omitted]

**4..0 Summary and Conclusions**

We report the expected atmospheric mixing times in $\alpha$-pinene SOA for atmospheric temperature and RH data, based on a parameterization developed using laboratory viscosity data of high mass loading SOA (520-1400 µg m$^{-3}$).  A parameterization for viscosity as a function of temperature and RH was developed for $\alpha$-pinene SOA based on room-temperature and low-temperature viscosity data of $\alpha$-pinene SOA generated in the laboratory using mass concentrations of ~1000 µg m$^{-3}$. We focused on ~1000 µg m$^{-3}$ because low-temperature and room-temperature viscosity measurements have been carried out using this mass concentration. Based on this parameterization, as well as RH and temperatures in the PBL, the mixing times within $\alpha$-pinene SOA areis < 1 h for 98.5 % and 99.9 % of the occurrences in the PBL during January and July, respectively, when monthly average total organic aerosol concentrations arewere > 0.5 µg m$^{-3}$ at the surface.It was determined that 98.5% and 99.9% of locations with significant SOA concentrations, for January and July respectively, will have rapid mixing times. Also based on this parameterization, 91.2 % and 97.5 % of the locations for January and July, respectively, have a mixing time < 0.1 h for conditions at the top of the PBL when monthly averaged total organic aerosol surface concentrations are > 0.5 µg m$^{-3}$.

As a starting point to quantify how often mixing times of organic molecules are < 1 h within α-pinene SOA generated using low mass concentrations, we developed a temperature-independent parameterization using the room-temperature viscosity data for α-pinene SOA from Zhang et al. (2015). Zhang et al. (2015) measured the viscosity of α-pinene SOA generated using a mass concentration of ~70 µg m$^{-3}$. Based on this temperature-independent parameterization, mixing times within α-pinene SOA are < 1 h for 45 and 38 % of the occurrences in the PBL during January and July, respectively, when monthly average total organic aerosol concentrations are > 0.5 µg m$^{-3}$ at the surface. However, several caveats need to be emphasized for these results. Most important, the results were based on room-temperature viscosity data only and the mixing times were calculated using the Stokes-Einstein relation, which can underpredict diffusion coefficients of organic molecules, and hence overpredict mixing times, when the viscosity of the matrix is high.

As a starting point to quantify how often mixing times of organic molecules are < 1 h within anthropogenic SOA, a parameterization for viscosity as a function of temperature and RH was developed using sucrose-water viscosity data. Based on this parameterization and assuming sucrose is a good proxy for anthropogenic SOA, 70 % and 83 % of the mixing times within anthropogenic SOA in the PBL are < 1 h for January and July, respectively, when SOA concentrations are significant (total organic aerosol concentration > 0.5 µg m$^{-3}$ at the surface). These percentages for anthropogenic SOA are likely lower limits since studies have shown that the Stokes-Einstein relation (which is used here to calculate diffusion coefficients of organic molecules from viscosities) can underpredict diffusion coefficients of organic molecules in sucrose-water mixtures by at least a factor of 10 to 100 at viscosities ≥ 10$^6$ Pa s (Chenyakin et al., 2017; Price et al., 2016).

To improve the predictions presented above the following are needed: 1) viscosities as a function of temperature and RH for α-pinene SOA and anthropogenic SOA generated using low mass concentrations and 2) studies that quantify the accuracy of the Stokes-Einstein equation for predicting diffusion coefficients in SOA. Studies that explore further the effect of oxidation level, oxidation type, and gas-phase precursor on viscosity and diffusion within biogenic and anthropogenic SOA would also be beneficial.

~~However, measurements have indicated that viscosities, and thus mixing times, increase with decreasing mass loading. We are left to conclude that additional studies are needed to fully understand the impact of mass loading on the viscosity and mixing times in SOA particles. Specific experiments that would be helpful are viscosity measurements at atmospherically relevant mass loadings at room-temperature and low temperatures across a range of RHs. As well, diffusion measurements in the SOA particles to determine the break-down of the Stokes-Einstein equation in the high viscosity SOA particles. 
[revised manuscript text omitted]
 the average afternoon surface temperatures mentioned above. The vertical profiles of RH were calculated using the average afternoon surface RHs mentioned above, the vertical profiles of temperature (calculated with the dry adiabatic lapse rate), and assuming the mixing ratio of water is independent of height in the PBL.

15  For the calculations of RH as a function of altitude, the water vapor pressure and water saturated vapour pressure were needed as a function of altitude. The water vapor pressure as a

20  function of altitude was determined by multiplying the mixing ratio of water by the atmospheric pressure. calculated using the following equation (Seinfeld and Pandis, 2006):

$$P(z) = P_0 \exp(-\frac{Mgz}{kT})$$

(S1)

25  where $P_0$ is the standard pressure at sea level (101325 Pa), M is the molecular mass of the air (28.8 g/mol), g is the acceleration due to gravity (9.81m s$^{-2}$), z is the altitude in metres, k is the Boltzmann constant and T is the temperature in Kelvin. The water saturated vapour pressure was calculated as a function of attitude using the Antoine equation (National Insititute of Standards and Technology, 2016):

30  $$log_{10}(P) = A - (\frac{B}{T+C})$$   (S2)

where P is the pressure, A=4.6543, B=1435.264, C=-64.848 and T is the temperature in Kelvin. The values for A, B and C were based on the NIST values for water, which are valid for temperatures between 256 and 373 K (National Insititute of Standards and Technology, 2016).

In Fig. 5, the temperature and RH were plotted until the RH reached 100 %. The height at which RH reached 100 % was only slightly lower than the average height of the planetary boundary layer predicted by GEOS-5 meteorology data for the driest month of the year and for the afternoon (13:00-15:00, local time) above Hyytiälä and the Amazon. For Hyytiälä, 100 % RH was reached at 1605 m, while GEOS-5 predicted an average height of the PBL of 1667 m for this location and time. For the Amazon, 100 % RH was reached at 882 m, while GEOS-5 predicted an average height of the PBL of 1249 m for this location and time. When predicting the height of the PBL using GEOS-5 meteorology, we ran GEOS-Chem at a horizontal grid resolution of 2° latitude by 2.5° longitude rather 4° latitude by 5° longitude to provide a better approximation to these single locations.

**S2. Parametrization for the viscosity of sucrose particles as a function of temperature and RH**

We developed a parameterization for viscosity of sucrose particles as function of temperature and RH by fitting the viscosity data listed in Table  S7 to the following equation:

$$\log(\eta) = 12 - \frac{C_1 * (T - \frac{w_{Suc}T_{gSuc} + w_{H2O}T_{gH2O}k_{GT}}{w_{Suc} + w_{H2O}k_{GT}})}{C_2 + (T - \frac{w_{Suc}T_{gSuc} + w_{H2O}T_{gH2O}k_{GT}}{w_{Suc} + w_{H2O}k_{GT}})} \qquad (\text{S3})$$

where $C_1$ and $C_2$ are constants, $k_{GT}$ is the Gordon-Taylor fitting parameter, $T_{gSuc}$ and $T_{gH2O}$ are the glass transition temperatures of dry sucrose and water and $w_{Suc}$ and $w_{H2O}$ are the weight fractions of the dry sucrose and water in the particles. The weight fractions of dry sucrose and water in the particles were determined from the RH using the following equation (Zobrist et al., 2011):

$$\frac{RH}{100} = \frac{1 + aw_{Suc}}{1 + bw_{Suc} + cw_{Suc}^2} + (T - T^\theta)(dw_{Suc} + ew_{Suc}^2 + fw_{Suc}^3 + gw_{Suc}^4) \qquad (\text{S4})$$

[revised manuscript text omitted]

[a] Grayson et al. (2016) reported upper and lower limits to the viscosity (i.e. range) at each specified RH.  To simplify the fitting procedure, we used the midpoints of the viscosities from Grayson et al (2016).

[b] Grayson et al. (2016) measured the viscosity under dry conditions (RH of < 0.5 % based on measurements).  When developing the parameterization, we used a value of 0 % RH.

[c] Grayson et al. (2016) carried out experiments at room temperature (293 K-295 K).  We used the midpoint of the temperature (294 K) when developing the viscosity parameterization for α-pinene SOA.

~~[d] Zhang et al. (2015) reported 36 measurements of viscosity over the range of 0 to 60%.  For the fitting procedure, we binned their data by relative humidity and used the average viscosity and relative humidity in each bin.  The width of each bin was approximately 10% RH.  This binning procedure was carried out to give the data from Grayson et al. (2016) and Zhang et al. (2015) similar weights, since both were carried out at room temperature and over a similar RH range.~~

**Table S2.** Low-temperature α-pinene SOA viscosity data from Järvinen et al. (2016). Viscosity  data corresponds to SOA generated with a mass concentration of 707-1414 μg m⁻³.

| Reference | Viscosity (Pa s) | RH (%) | Temperature (K) |
|---|---|---|---|
| Järvinen et al. (2016) | 1x10⁷ | [a]Range=22.9-36.3, midpoint=29.6 | 263.3 |
| | | [a]Range=30.5-37.3, midpoint=33.9 | 262.9 |
| | | [a]Range=40.5-46.0, midpoint=43.3 | 253.3 |
| | | [a]Range=44.0-49.8, midpoint=46.9 | 252.9 |
| | | [a]Range=55.0-63.4, midpoint=59.2 | 243.3 |
| | | [a]Range=68.6-80.1, midpoint=74.4 | 235.5 |

[a] Järvinen et al (2016) reported upper and lower limits to the RH for a specific temperature and viscosity.  To simplify fitting, we used the midpoint of the RH range.

Table S3. Liquid water viscosity data from Crittenden et al. (2012).

Field Code Changed

| Reference | Viscosity (Pa s) | RH (%) | Temperature (K) |
|---|---|---|---|
| Crittenden et al. (2012) | [a]1.002x10$^{-3}$ | | 293 |
| | [a]1.139 x10$^{-3}$ | | 288 |
| | [a]1.307 x10$^{-3}$ | 100 | 283 |
| | [a]1.518 x10$^{-3}$ | | 278 |
| | [a]1.781 x10$^{-3}$ | | 273 |

Field Code Changed

[a] The viscosit values in Crittenden et al. (2012) were reported to 4 significant digits.

Field Code Changed

Table S4. Initial guess parameters and fitting parameters used in Eq. (1) to predict the viscosity of α-pinene SOA as a function of temperature and RH.  The fitting parameters were obtained by fitting Eq. (1) to the viscosity data listed in Tables S1-S3.

| Parameter | Guess Value | Fitting Value |
|---|---|---|
| $C_1$ | 19 | 131 |
| $C_2$ | 50 K | 1165 K |
| $K_{GT}$ | 2.5 | 3.934 |
| Tg$_{SOA}$ | 250 K | 236.8 K |

Table S5. Room temperature α-pinene SOA viscosity data from Zhang et al. (2015).  Viscosity data corresponds to SOA generated with a mass concentration of ~70 µg m$^{-3}$.

Field Code Changed

| Reference | Viscosity (Pa s) | RH (%) | Temperature (K) |
|---|---|---|---|
| Zhang et al. (2015)[a] | 2.3 x10$^8$ | 5.2 | 293 |
| | 1.3 x10$^8$ | 13.8 | 293 |
| | 3.2 x10$^7$ | 22.9 | 293 |
| | 1.4 x10$^7$ | 36.7 | 293 |
| | 6.0 x10$^6$ | 44.3 | 293 |
| | 5.1 x10$^6$ | 54.3 | 293 |

Field Code Changed

[a]Zhang et al. (2015) reported 36 measurements of viscosity over the range of  0 to 60 %.  For the fitting procedure, we binned their data by relative humidity and used the average viscosity and relative humidity in each bin.  The width of each bin was approximately 10 % RH.

Field Code Changed

**Table S6.** Initial guess parameters and fitting parameters used in Eq. (1) to develop a temperature-independent parameterization for viscosity of  $\alpha$-pinene SOA. The fitting parameters were obtained by fitting Eq. (1) to the room-temperature viscosity data from Zhang et al. ( 2015) and Crittenden et al. (2012), but with the temperature (T) in Eq. (1) replaced by 293 K.

| Parameter | Guess Value | Fitting Value |
|---|---|---|
| $C_1$ | 19 | 18.73 |
| $C_2$ | 50 K | 29.25 |
| $K_{GT}$ | 2.5 | 0.1628 |
| $Tg_{SOA}$ | 250 K | 285.9 K |

**Table S7.** Literature viscosity data used to create a parameterization for the viscosity of sucrose particles as a function of temperature and RH.

| System | Viscosity Range (Pa s) | RH (%) | Temperature (K) | Reference |
|---|---|---|---|---|
| Water | $1.002 \times 10^{-3}$ to $1.781 \times 10^{-3}$ | 100 | 275-293 | Crittenden et al. (2012) |
| Sucrose-water | $3.19 \times 10^{-3}$ to $4.82 \times 10^{-1}$ | 96.2-80 | 293 | Swindells et al. (1958) |
| | $6.73 \times 10^{-1}$ to $1.10 \times 10^{3}$ | 80-56.6 | | Quintas et al. (2006) |
| | $1.97 \times 10^{-3}$ to $5.67 \times 10^{-2}$ | 99.4-88 | | Perry and Green (2008) |
| | $1.25 \times 10^{-3}$ to $8.30 \times 10^{-2}$ | 99.99-87.96 | | Migliori et al. (2007) |
| | $1.26 \times 10^{-3}$ to $7.65 \times 10^{-2}$ | 99.89-87.98 | | Telis et al. (2007) |
| | $1.03 \times 10^{-3}$ to $5.81 \times 10^{-2}$ | 100-87.98 | | Forst et al. (2002) |

| | | | Power and Reid (2014) |
|---|---|---|---|
| $3\times10^{-2}$ to $6.71\times10^{8}$ | 92-28 | | Power and Reid (2014) |
| $1\times10^{12}$ | 48.53-25.88 | 255-295 (5 degree increments)[a] | Zobrist et al. (2008) |

[a] Zobrist et al. (2008) reported glass transition temperatures as a function of water activity for the range of 160 K to 300 K. These glass transition temperatures were based on glass transition temperature measurements in the range of 180240 K to 240180 K, water activity measurements, and the Gordon-Taylor equation. To develop our parameterization, we used their glass transition temperatures over the range of 255 K to 295 K from their Fig. 5b, recorded in 5 K increments.

**Table S6S8.** Parameters from Zobrist et al. (2011) used in Eq. (S46) to predict the weight fractions of sucrose and water in particles as a function of relative humidity.

| Parameter | Value | Parameter | Value |
|---|---|---|---|
| a | -1 | e | -0.005151 |
| b | -0.99721 | f | 0.009607 |
| c | 0.13599 | g | -0.006142 |
| d | 0.001688 | $T^o$ | 298 K |

**Table S7S9.** Fitting parameters used in Eq. (S35) to predict the viscosity of sucrose particles as a function of temperature and RH. These parameters were obtained by fitting Eq. (S35) to the viscosity data listed in Table S5 S76 as well as the guess values in the table.

| Parameter | Guess Value | Fitting Value |
|---|---|---|
| $C_1$ | 19 | 20.06 |
| $C_2$ | 50 K | 55.58 K |
| $K_{GT}$ | 4.74 | 4.531 |
| $Tg_{SOA}$ | 336 K | 324.5 K |

**Figures**

[Figure]

**Figure S1.** Monthly averaged Rrelative humidity and temperature at the surface. Panels (a) and (c) correspond to January and panels (b) and (d) correspond to July.

[Figure]

**Figure S2.** Monthly aAverage relative humidity and temperature at the top of the planetary boundary layer. Panels (a) and (c) correspond to January, and panels (b) and (d) correspond to July.

[Figure]

**Figure S3.** Normalized frequency distributions of mixing times within 500 nm α-pinene SOA  in the planetary boundary layer (PBL). Black symbols correspond to January and red symbols corresponds to July. Frequency counts in the PBL were only included for the conditions where the mass concentration of total organic aerosol was > 0.5 μg m$^{-3}$ at the surface. The viscosity parameterization used to calculate mixing times was based on α-pinene SOA generated using mass concentrations of ~1000 μg m$^{-3}$.

[Figure]

**Figure S4.** Normalized frequency distributions of mixing times within α-pinene SOA in the planetary boundary layer (PBL) in January for the parameterizations generated using the upper limit of the viscosity data from Grayson et al. ( 2016) and the upper RH limit from Järvinen et al. (2016). Blue symbols correspond to January and red symbols correspond to July. Frequency counts in the PBL were only included for the conditions where the mass concentration of total organic aerosol was > 0.5 μg m$^{-3}$ at the surface.

[Figure]

**Figure S5.** Viscosities of α-pinene particles as a fucntion of  RH from Zhang et al. ( 2015) and Grayson et al. ( 2016) as well as the viscosity of water at room temperature from Crittenden et al. ( 2012). The viscosity data from Grayson et al. (2016) correspond to a SOA mass concentration of 520 μg m$^{-3}$, and the viscosity data from Zhang et al. (2015) correspond to a SOA mass concentration of 70 μg m$^{-3}$.

[Figure]

**Figure S1S6.** Viscosities of different proxies of anthropogenic SOA as a function of RH. Data for toluene SOA taken from Song et al. (2016). The data for sucrose-water mixtures was taken from Swindells (1958), Quintas et al. (2006), Telis et al. (2007), Forst et al. (2002), Migliori et al. (2007), Perry and Green (2008), and Power and Reid (2014).

[Figure]

**Figure S2S7.** Panel A: Parameterization (contours) for the viscosity of sucrose particles (as surrogates of  anthropogenic SOA) as a function of temperature and RH and measured viscosities used to construct the parameterization (symbols). The measured viscosities are listed in Table S5S76. Panel B: Mixing times (color scale) for organic molecules within 200 nm sucrose particles as a function of temperature and RH. Mixing times were calculated from the viscosity parameterization (Panel A) and Eq. (53) and (64) in the main text.

[Figure]

**Figure** S3S8. Six-hour normalized frequency counts of temperature and RH in the planetary boundary layer (color scale) together with the mixing times for organic molecules within 200 nm sucrose particles (as surrogates of toluene anthropogenic SOA) (contours). Panels A and B show the conditions for January and July, respectively. Mixing times (contours) are reported in hours.

Frequency counts in the PBL were only included for the conditions when the mass concentration of total organic aerosol was > 0.5 μg/m³ at the surface.

[Figure]

5 **Figure S4S9**. Normalized frequency distributions of mixing times within sucrose particles (as surrogates for  anthropogenic SOA) in the planetary boundary layer. Red symbols correspond to January and blue symbols correspond to July. Frequency counts in the PBL were only included for the conditions where the mass concentration of total organic aerosol was > 0.5 μg m⁻³ at the surface. The relatively large frequency count at 5x10⁵ h is because all cases 10 that had a viscosity greater than 10¹² Pa s were assigned a value of 10¹² Pa s. For additional details see Section S2.

[Figure]

[Figure]

Commented [AB6]: This figure should be modified like the figure in the main text. I.e. only show colors for conditions where OA < 0.5 micrograms/m^3

**Figure** S10. Mixing times of organic molecules within 200 nm sucrose particles (as surrogates of  anthropogenic SOA) at the top of the planetary boundary layer as a function of latitude and longitude. The color scale represents mixing times. Mixing times are only shown for locations with total organic aerosol concentrations > 0.5 ug m$^{-3}$ at the surface.

mixing times and the yellow contours illustrate when the concentration of total organic aerosol is > 0.5 ug m⁻³ at the surface. Panels A and B correspond to January and July, respectively.

---

## Author Response (AR2)

Professor Nga Lee (Sally) Ng
Co-Editor of Atmospheric Chemistry and Physics

5  Dear Sally,

Listed below are our responses to the comments from the reviewer.  For clarity and visual
distinction, the referee comments are listed here in black and are preceded by bracketed,
italicized numbers (e.g. *[1]*). Authors' responses are in red below each referee statement
10  with matching numbers (e.g. *[A1]*).  I would like to thank you very much for being the co-
editor of our manuscript!

Sincerely,

15  Allan

Anonymous Referee #1
20  It is clear that the authors have done a thorough job with their revisions, and what they
have done is much clearer now. It is, however, unfortunate that their conclusions now
become much less solid w.r.t. the biogenic SOA case, in particular. In essence, for
biogenic SOA they either conclude that it is always well mixed (one limit) or often not well-
mixed (the other limit). This speaks more to the constraints on the available data, and the
25  inconsistencies in the literature, than it does to the specific work here.

*[1]* As such, I suggest that the authors update their abstract to more directly point out what
they now state in their revised manuscript, namely that "Considering these caveats, we
are unable to make strong conclusions about how often mixing times of organic molecules
30  are < 1 h within a-pinene SOA generated at low mass concentrations."

*[A1]* The following sentence has been added to the abstract:

"However, associated with these conclusions are several caveats, and due to these
35  caveats, we are unable to make strong conclusions about how often mixing times of
organic molecules are < 1 h within α-pinene SOA generated using low, atmospherically
relevant,  mass concentrations"

*[2]* But, I actually think it goes beyond this because what the goal is (presumably) is not
40  to assess global variability in mixing times at very high and moderate OA concentrations,
but to assess what the real atmosphere does. Thus, I suggest the authors more directly
say that the difference between the parameterizations means that they also cannot make
"strong conclusions" regarding the mixing timescales of biogenic SOA under ambient
concentrations. This should be stated in the main text and in the abstract in some manner.
45

*[A2]* To address the referee's comment, in the abstract and in the main text, we have changed "low mass concentrations" to "low, atmospherically relevant, mass concentration". This should make it clear that we are unable to make strong conclusions for ambient concentrations.

*[3]* Also, I suggest that they change "low mass concentrations" on P7/L7 to "low, atmospherically relevant mass concentrations".

*[A3]* This change has been made as suggested.

[revised manuscript text omitted]
 the average afternoon surface temperatures mentioned above. The vertical profiles of RH were calculated using the average afternoon surface RHs mentioned above, the vertical profiles of temperature (calculated with the dry adiabatic lapse rate), and assuming the mixing ratio of water is independent of height in the PBL. For the calculations of RH as a function of altitude, the water vapor pressure and water saturated vapour pressure were needed as a function of altitude. The water vapor pressure as a function of altitude was determined by multiplying the mixing ratio of water by the atmospheric pressure, calculated using the following equation (Seinfeld and Pandis, 2006):

$$P(z) = P_0 \exp(-\frac{Mgz}{kT}) \tag{S1}$$

where $P_0$ is the standard pressure at sea level (101325 Pa), M is the molecular mass of the air (28.8 g/mol), g is the acceleration due to gravity (9.81m s$^{-2}$), z is the altitude in metres, k is the Boltzmann constant and T is the temperature in Kelvin. The water saturated vapour pressure was calculated as a function of attitude using the Antoine equation (National Insititute of Standards and Technology, 2016):

$$log_{10}(P) = A - (\frac{B}{T+C}) \tag{S2}$$

where P is the pressure, A=4.6543, B=1435.264, C=-64.848 and T is the temperature in Kelvin. The values for A, B and C were based on the NIST values for water, which are valid for temperatures between 256 and 373 K (National Insititute of Standards and Technology, 2016).

In Fig. 5, the temperature and RH were plotted until the RH reached 100 %. The height at which RH reached 100 % was only slightly lower than the average height of the planetary boundary layer

predicted by GEOS-5 meteorology data for the driest month of the year and for the afternoon (13:00-15:00, local time) above Hyytiälä and the Amazon. For Hyytiälä, 100 % RH was reached at 1605 m, while GEOS-5 predicted an average height of the PBL of 1667 m for this location and time. For the Amazon, 100 % RH was reached at 882 m, while GEOS-5 predicted an average height of the PBL of 1249 m for this location and time. When predicting the height of the PBL using GEOS-5 meteorology, we ran GEOS-Chem at a horizontal grid resolution of 2° latitude by 2.5° longitude rather 4° latitude by 5° longitude to provide a better approximation to these single locations.

**S2. Parametrization for the viscosity of sucrose particles as a function of temperature and RH**

We developed a parameterization for viscosity of sucrose particles as function of temperature and RH by fitting the viscosity data listed in Table S7 to the following equation:

$$\log(\eta) = 12 - \frac{C_1 * (T - \frac{w_{Suc}T_{gSuc} + w_{H2O}T_{gH2O}k_{GT}}{w_{Suc} + w_{H2O}k_{GT}})}{C_2 + (T - \frac{w_{Suc}T_{gSuc} + w_{H2O}T_{gH2O}k_{GT}}{w_{Suc} + w_{H2O}k_{GT}})} \tag{S3}$$

where $C_1$ and $C_2$ are constants, $k_{GT}$ is the Gordon-Taylor fitting parameter, $T_{gSuc}$ and $T_{gH2O}$ are the glass transition temperatures of dry sucrose and water and $w_{Suc}$ and $w_{H2O}$ are the weight fractions of the dry sucrose and water in the particles. The weight fractions of dry sucrose and water in the particles were determined from the RH using the following equation (Zobrist et al., 2011):

$$\frac{RH}{100} = \frac{1 + aw_{Suc}}{1 + bw_{Suc} + cw_{Suc}^2} + (T - T^\theta)(dw_{Suc} + ew_{Suc}^2 + fw_{Suc}^3 + gw_{Suc}^4) \tag{S4}$$

[revised manuscript text omitted]

[a]Zhang et al. (2015) reported 36 measurements of viscosity over the range of 0 to 60 %. For the fitting procedure, we binned their data by relative humidity and used the average viscosity and relative humidity in each bin. The width of each bin was approximately 10 % RH.

**Table S6.** Initial guess parameters and fitting parameters used in Eq. (1) to develop a temperature-independent parameterization for viscosity of α-pinene SOA. The fitting parameters were obtained by fitting Eq. (1) to the room-temperature viscosity data from Zhang et al. (2015) and Crittenden et al. (2012), but with the temperature (T) in Eq. (1) replaced by 293 K.

| Parameter | Guess Value | Fitting Value |
|---|---|---|
| $C_1$ | 19 | 18.73 |
| $C_2$ | 50 K | 29.25 |
| $K_{GT}$ | 2.5 | 0.1628 |
| $Tg_{SOA}$ | 250 K | 285.9 K |

**Table S7.** Literature viscosity data used to create a parameterization for the viscosity of sucrose particles as a function of temperature and RH.

| System | Viscosity Range (Pa s) | RH (%) | Temperature (K) | Reference |
|---|---|---|---|---|
| Water | $1.002 \times 10^{-3}$ to $1.781 \times 10^{-3}$ | 100 | 275-293 | Crittenden et al. (2012) |
| Sucrose-water | $3.19 \times 10^{-3}$ to $4.82 \times 10^{-1}$ | 96.2-80 | | Swindells et al. (1958) |
| | $6.73 \times 10^{-1}$ to $1.10 \times 10^3$ | 80-56.6 | | Quintas et al. (2006) |

| | | | | |
|---|---|---|---|---|
| | 1.97x10$^{-3}$ to 5.67x10$^{-2}$ | 99.4-88 | 293 | Perry and Green (2008) |
| | 1.25x10$^{-3}$ to 8.30x10$^{-2}$ | 99.99-87.96 | | Migliori et al. (2007) |
| | 1.26x10$^{-3}$ to 7.65x10$^{-2}$ | 99.89-87.98 | | Telis et al. (2007) |
| | 1.03x10$^{-3}$ to 5.81x10$^{-2}$ | 100-87.98 | | Forst et al. (2002) |
| | 3x10$^{-2}$ to 6.71x10$^{8}$ | 92-28 | | Power and Reid (2014) |
| | 1x10$^{12}$ | 48.53-25.88 | 255-295 (5 degree increments)[a] | Zobrist et al. (2008) |

[a] Zobrist et al. (2008) reported glass transition temperatures as a function of water activity for the range of 160 K to 300 K. These glass transition temperatures were based on glass transition temperature measurements in the range of 180 K to 240 K, water activity measurements, and the Gordon-Taylor equation. To develop our parameterization, we used their glass transition temperatures over the range of 255 K to 295 K from their Fig. 5b, recorded in 5 K increments.

**Table S8.** Parameters from Zobrist et al. (2011) used in Eq. (S4) to predict the weight fractions of sucrose and water in particles as a function of relative humidity.

| Parameter | Value | Parameter | Value |
|---|---|---|---|
| a | -1 | e | -0.005151 |
| b | -0.99721 | f | 0.009607 |
| c | 0.13599 | g | -0.006142 |
| d | 0.001688 | T$^{e}$ | 298 K |

**Table S9.** Fitting parameters used in Eq. (S3) to predict the viscosity of sucrose particles as a function of temperature and RH. These parameters were obtained by fitting Eq. (S3) to the viscosity data listed in Table S7 as well as the guess values in the table.

| Parameter | Guess Value | Fitting Value |
|---|---|---|
| $C_1$ | 19 | 20.06 |
| $C_2$ | 50 K | 55.58 K |
| $K_{GT}$ | 4.74 | 4.531 |
| $Tg_{SOA}$ | 336 K | 324.5 K |

**Figures**

[Figure]

**Figure S1.** Monthly average relative humidity and temperature at the surface. Panels (a) and (c) correspond to January and panels (b) and (d) correspond to July.

[Figure]

**Figure S2.** Monthly average relative humidity and temperature at the top of the planetary boundary layer. Panels (a) and (c) correspond to January, and panels (b) and (d) correspond to July.

[Figure]

**Figure S3.** Normalized frequency distributions of mixing times within 500 nm α-pinene SOA in the planetary boundary layer (PBL). Black symbols correspond to January and red symbols corresponds to July. Frequency counts in the PBL were only included for the conditions where the mass concentration of total organic aerosol was > 0.5 μg m$^{-3}$ at the surface. The viscosity parameterization used to calculate mixing times was based on α-pinene SOA generated using mass concentrations of ~1000 μg m$^{-3}$.

[Figure]

**Figure S4.** Normalized frequency distributions of mixing times within α-pinene SOA in the planetary boundary layer (PBL) in January for the parameterizations generated using the upper limit of the viscosity data from Grayson et al. (2016) and the upper RH limit from Järvinen et al. (2016). Blue symbols correspond to January and red symbols correspond to July. Frequency counts in the PBL were only included for the conditions where the mass concentration of total organic aerosol was > 0.5 μg m$^{-3}$ at the surface.

[Figure]

**Figure S5.** Viscosities of α-pinene particles as a fucntion of RH from Zhang et al. (2015) and Grayson et al. (2016) as well as the viscosity of water at room temperature from Crittenden et al. (2012). The viscosity data from Grayson et al. (2016) correspond to a SOA mass concentration of 520 μg m$^{-3}$, and the viscosity data from Zhang et al. (2015) correspond to a SOA mass concentration of 70 μg m$^{-3}$.

[Figure]

**Figure S6.** Viscosities of different proxies of anthropogenic SOA as a function of RH. Data for toluene SOA taken from Song et al. (2016). The data for sucrose-water mixtures was taken from Swindells (1958), Quintas et al. (2006), Telis et al. (2007), Forst et al. (2002), Migliori et al. (2007), Perry and Green (2008), and Power and Reid (2014).

[Figure]

**Figure S7.** Panel A: Parameterization (contours) for the viscosity of sucrose particles (as surrogates of anthropogenic SOA) as a function of temperature and RH and measured viscosities used to construct the parameterization (symbols). The measured viscosities are listed in Table S7.

5   Panel B: Mixing times (color scale) for organic molecules within 200 nm sucrose particles as a function of temperature and RH. Mixing times were calculated from the viscosity parameterization (Panel A) and Eq. (5) and (6) in the main text.

[Figure]

**Figure S8.** Six-hour normalized frequency counts of temperature and RH in the planetary boundary layer (color scale) together with the mixing times for organic molecules within 200 nm sucrose particles (as surrogates of anthropogenic SOA) (contours). Panels A and B show the conditions for January and July, respectively. Mixing times (contours) are reported in hours.

Frequency counts in the PBL were only included for the conditions when the mass concentration of total organic aerosol was > 0.5 μg/m$^3$ at the surface.

[Figure]

**Figure S9.** Normalized frequency distributions of mixing times within sucrose particles (as surrogates for anthropogenic SOA) in the planetary boundary layer. Red symbols correspond to January and blue symbols correspond to July. Frequency counts in the PBL were only included for the conditions where the mass concentration of total organic aerosol was > 0.5 μg m$^{-3}$ at the surface. The relatively large frequency count at 5x10$^5$ h is because all cases that had a viscosity greater than 10$^{12}$ Pa s were assigned a value of 10$^{12}$ Pa s. For additional details see Section S2.

[Figure]

**Figure S10.** Mixing times of organic molecules within 200 nm sucrose particles (as surrogates of anthropogenic SOA) at the top of the planetary boundary layer as a function of latitude and longitude. The color scale represents mixing times. Mixing times are only shown for locations with total organic aerosol concentrations > 0.5 ug m$^{-3}$ at the surface. Panels A and B correspond to January and July, respectively.